# Secretory GFP reconstitution labeling of neighboring cells interrogates cell–cell interactions in metastatic niches

Misa Minegishi[1,2], Takahiro Kuchimaru [2,3,4,5,15] ✉, Kaori Nishikawa[2], Takayuki Isagawa[4,5], Satoshi Iwano[6,7], Kei Iida [8], Hiromasa Hara[4], Shizuka Miura[9], Marika Sato[10], Shigeaki Watanabe[10], Akifumi Shiomi [2], Yo Mabuchi [11,12], Hiroshi Hamana[13], Hiroyuki Kishi [13], Tatsuyuki Sato [4], Daigo Sawaki[4,14], Shigeru Sato[4], Yutaka Hanazono[4], Atsushi Suzuki [9], Takahide Kohro[5], Tetsuya Kadonosono[1], Tomomi Shimogori[6], Atsushi Miyawaki [6], Norihiko Takeda[4], Hirofumi Shintaku [2,15] ✉, Shinae Kizaka-Kondoh [1] & Satoshi Nishimura[4]

Cancer cells inevitably interact with neighboring host tissue-resident cells during the process of metastatic colonization, establishing a metastatic niche to fuel their survival, growth, and invasion. However, the underlying mechanisms in the metastatic niche are yet to be fully elucidated owing to the lack of methodologies for comprehensively studying the mechanisms of cell–cell interactions in the niche. Here, we improve a split green fluorescent protein (GFP)-based genetically encoded system to develop secretory glycosylphosphatidylinositol-anchored reconstitution-activated proteins to highlight intercellular connections (sGRAPHIC) for efficient fluorescent labeling of tissue-resident cells that neighbor on and putatively interact with cancer cells in deep tissues. The sGRAPHIC system enables the isolation of metastatic niche-associated tissue-resident cells for their characterization using a single-cell RNA sequencing platform. We use this sGRAPHIC-leveraged transcriptomic platform to uncover gene expression patterns in metastatic niche-associated hepatocytes in a murine model of liver metastasis. Among the marker genes of metastatic niche-associated hepatocytes, we identify Lgals3, encoding galectin-3, as a potential pro-metastatic factor that accelerates metastatic growth and invasion.

Cellular functions in living tissues are often dictated by local cell–cell interactions via diverse juxtacrine and paracrine factors. For instance, in tissue homeostasis, functions of hematopoietic stem cells are regulated by interactions with neighboring mesenchymal stromal cells in the bone marrow niche[1,2]. Similarly, cancer cell–tissue-resident cell interactions play critical roles in driving cancer malignancies[3]. During metastasis, cancer cells form metastatic niches to gain benefits for their growth through interactions with neighboring tissue-resident cells during metastatic colonization processes[4–6]. Although understanding the cell–cell interactions in the metastatic niche has been a long-standing goal for preventing metastatic disease, few studies have successfully detailed cell–cell interactions in the niche owing to the lack of methodologies allowing for comprehensive investigation of metastatic niches[7]. The investigation of cell–cell interactions in living

tissues has still relied on classical immunohistological analysis of fixed tissue sections[7,8]. Optical labeling of cell−cell interactions in living tissues is an emerging approach to achieve live cell-based analysis of interacting cells isolated from complex multicellular organization. Immune cell interactions in the murine lymphatic tissue have been successfully characterized through in vivo optical labeling with photoactivatable green fluorescent protein (GFP)[9]. More recently, the Cherry-niche system was used to elegantly label and isolate tissue-resident cells involved in the murine lung metastatic niche by using cell-permeable fluorescent proteins secreted from neighboring cancer cells[10]. These successful attempts further demonstrated that optical-labeling approaches in living tissues could be combined with single-cell transcriptomics of isolated cells involved in cell−cell interactions[9,10], thereby surpassing the classical histological analysis for testing unbiased hypotheses on the molecular machinery underlying the cell−cell interactions. Therefore, expanding the toolbox of the optical-labeling approaches holds great promise for interrogating cell−cell interactions in living tissues; however, it remains challenging to develop genetically encoded optical-labeling tools for harnessing high labeling efficiency of a variety of cell types.

In this work, we developed a genetically encoded optical-labeling system, named secretory glycosylphosphatidylinositol-anchored reconstitution-activated proteins to highlight intercellular connections (sGRAPHIC), through optimization of the previous our GRAPHIC system[11]. sGRAPHIC allows efficient labeling of various cancer cell−tissue-resident cell interactions through intercellular reconstitution of split-GFP fragments. We used sGRAPHIC to successfully label hepatocytes interacting proximally with cancer cells in liver metastatic niches in mice. Furthermore, coupling sGRAPHIC and flow sorting facilitated the single-cell RNA sequencing (scRNA-seq) of tissue-resident cells involved in metastatic niches. This sGRAPHIC-leveraged transcriptomic platform revealed that metastatic niche-associated hepatocytes (MAHs) fuel metastatic progression via secreted galectin-3. Overall, our results suggest that the sGRAPHIC is a powerful strategy for interrogating cell−cell interactions in metastatic niches, aiding in accelerating our understanding of the disease progression.

## Results

### Development of a fluorescent labeling system for capturing broad cell−cell interactions

Previous GRAPHIC systems employed GPI-anchored split-GFP fragments to specifically detect physical cell−cell contacts[11,12]. We tested the applicability of the GRAPHIC system in cancer cell lines by genetic transduction of C-GRAPHIC (C-GR) or N-GRAPHIC (N-GR) reporter. These reporters express cell membrane-anchored C- or N-terminal GFP fragments. To distinguish cells expressing N- or C-terminal GFP fragments by a fluorescent maker of the nucleus, the reporters also encode fusion proteins of histone H2B protein, and red fluorescent protein mCherry or blue fluorescent protein Azurite (Supplementary Fig. 1a). The GRAPHIC system efficiently labeled cell−cell interactions in the epithelial cell line LLC-PK1 cells as we previously demonstrated[11], but the system was not functional in the cancer cell line HeLa cells (Supplementary Fig. 1b). We speculated that the inefficient fluorescence labeling of GRAPHIC in cancer cells was due to the unstable cell−cell adhesion of cancer cells. To achieve efficient optical labeling of cell−cell interactions involving cancer cells, sGRAPHIC was conceived using a combination of GPI-anchored N-terminal and secretory C-terminal GFP fragments (Fig. 1a). To develop sGRAPHIC, we firstly designed secretory N-GRAPHIC (sN-GR) and secretory C-GRAPHIC (sC-GR) reporters that extracellularly secrete N- or C-terminal GFP fragments (Supplementary Fig. 1c); these reporters reconstitute GFP with C-GRAPHIC (C-GR) or N-GRAPHIC (N-GR) reporter that is cell membrane-anchored C- or N-terminal GFP fragments. Both sN-GR and sC-GR successfully expressed GFP fragments and reconstituted GFP

when C-GR and N-GR, respectively, were co-expressed in the same cells (Supplementary Fig. 1d); however, when expressed in different cells and co-cultured, only the combination of sC-GR and N-GR efficiently reconstituted GFP in N-GR-expressing cells (Supplementary Fig. 1e). We speculated that this difference in GFP reconstitution was likely due to the secretion efficiency of the GFP fragments. We assessed the secretion efficiency of the GFP fragments by tagging them with the HiBiT system[13], which does not interfere with the GFP reconstitution capacities of both sN-GR and sC-GR (Supplementary Fig. 2a, b). Luminescence detection of HiBiT-tagged GFP fragments in cell culture supernatants indicated that the secretion efficiency of sC-GR was much higher than that of sN-GR (Supplementary Fig. 2c). On the basis of these results, we adopted a combination of sC-GR and N-GR for sGRAPHIC (Fig. 1a).

To test sGRAPHIC for neighboring cell labeling, we established HEK293T and HeLa cells stably expressing sGRAPHIC reporters. GFP was successfully reconstituted on neighboring HEK293T cells stably expressing N-GR (HEK293T/N-GR) when they were co-cultured with HKE293T cells stably expressing sC-GR (HEK293T/sC-GR) for 24 h regardless of the cell ratio (Fig. 1b and Supplementary Fig. 2d). Similarly, sGRAPHIC efficiently labeled neighboring cells with GFP in co-culture of HeLa cells stably expressing sGRAPHIC reporters (Supplementary Fig. 2e). Reconstituted GFPs were predominantly localized to the cell membrane as demonstrated by the loss of GFP signals after promiscuous digestive enzymatic treatments (Supplementary Fig. 3). As shown in Supplementary Fig. 2d, when HEK293T/N-GR cells and HEK293T/sC-GR cells are co-cultured subconfluently at a ratio of 1:1, the majority of HEK293T/N-GR cells were neighboring on HEK293T/sC-GR cells. Under the same co-culture conditions as Supplementary Fig. 2d, HEK293T/N-GR cells were efficiently labeled within 6 h, as determined by flow cytometry analysis (Fig. 1c). This result motivated us to compare the labeling efficiency of sGRAPHIC with the existing niche labeling tool by secreting fluorescent protein, Cherry-niche[10]. We established stable HEK293T cell lines that highly expressed Cherry-niche or H2B-Azurite (Supplementary Fig. 4a). After co-culturing these cell lines at a ratio of 1:1, flow cytometry analysis and confocal fluorescence observation showed that small populations of HEK293T/H2B-Azurite cells were labeled with mCherry (Fig. 1d and Supplementary Fig. 4b). Quantitative analysis revealed that the ratio of GFP-positive cells labeled using sGRAPHIC significantly surpassed the ratio of mCherry-positive cells labeled using Cherry-niche (Fig. 1e). These results demonstrated that sGRAPHIC is capable of efficiently labeling N-GR expressing cells through the reconstitution of GFP with secretory GFP fragments.

### sGRAPHIC labeling for a variety of cell−cell interactions

To demonstrate the versatility of sGRAPHIC labeling of various cancer cell−tissue-resident cell interactions, we established murine breast cancer cell line E0771 (E0771/sC-GR) and murine colon cancer cell line MC-38 stably expressing sC-GR (MC-38/sC-GR). We also established several murine tissue-resident cell lines stably expressing N-GR, including the fibroblast cell line NIH3T3 (NIH3T3/N-GR), the endothelial cell line MS1 (MS1/N-GR), the osteoblast cell line KUSA-A1 (KUSA-A1/N-GR), and the hepatocyte cell line AML12 (AML12/N-GR). When the N-GR expressing tissue-resident cell lines were co-cultured with cancer cells expressing sC-GR for 24 h at a ratio of 1:1, fluorescent confocal microscope observation and flow cytometry analysis confirmed that the tissue-resident cells were selectively labeled with GFP at high efficiency (Supplementary Figs. 5 and 6). Consistent with the flow cytometry analysis at 6 h in co-culture assessment (Fig. 1c), time-lapse fluorescent imaging revealed that GFP signals were detectable approximately 3 h after co-culturing cancer cells and fibroblasts (Fig. 2a and Supplementary Movies 1, 2). To observe the diffusion of sC-GR from the cancer cells, we co-cultured cancer cells and fibroblasts at a ratio of 1:50 for 24 h. This assay revealed that GFP-labeled fibroblasts

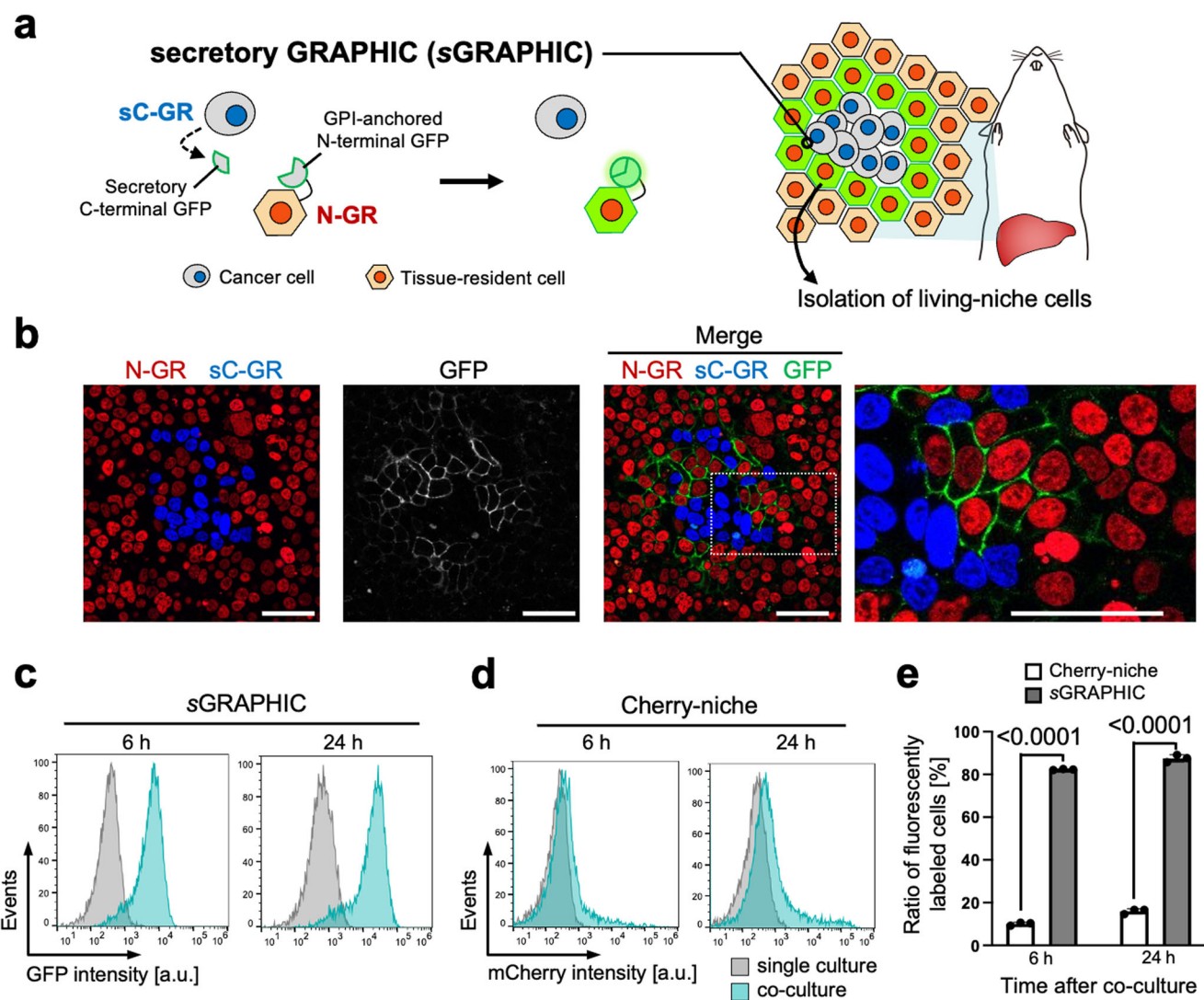

**Fig. 1 | Development of a system for secretory GFP reconstitution labeling of neighboring cells. a** Schematic of *s*GRAPHIC strategy for GFP labeling of metastatic niche cells. C-terminal GFP fragments are secreted from sC-GR expressing cells. The secreted C-terminal GFP fragments are reconstituted with N-terminal GFP fragments displayed on the plasma membrane of N-GR expressing cells. **b** *s*GRAPHIC labeling in HEK293T cells. *s*GRAPHIC specifically labeled N-GR expressing HEK293T cells (red nuclei) neighboring on sC-GR expressing HEK293T cells (blue nuclei). The white-rectangle area is enlarged in the right panel. Similar results were observed in multiple fields of view in independent duplicate experiments. Scale bars indicate 50 μm. **c** Flow cytometry analysis of *s*GRAPHIC labeling. HEK293T/N-GR cells and HEK293T/sC-GR cells were co-cultured in equal numbers for 6 or 24 h, and the GFP intensity of mCherry-positive cells was measured. Similar results were observed in independent triplicate experiments. Source data are provided as a Source Data file. **d** Flow cytometry analysis of Cherry-niche labeling. The same number of HEK293T/Cherry-niche cells and HEK293T/H2B-Azurite cells were co-cultured for 6 or 24 h, and the mCherry intensity of Azurite-positive cells was measured. Similar results were observed in independent triplicate experiments. Source data are provided as a Source Data file. **e** Labeling efficiency of *s*GRAPHIC and Cherry-niche in flow cytometry measurements. Data were statistically analyzed with two-tailed Student's t-test ($n = 3$ biologically independent samples). The *p*-values are indicated in the graph. Data are presented as mean values ± SEM. Source data are provided as a Source Data file.

were observed up to one- or two- cell layers away from the cancer cells (Fig. 2b), and a small number of GFP-positive fibroblasts were detected by flow cytometry (Supplementary Fig. 7). We also observed that reconstituted-GFP signals were undetectable for several hours after the cancer cells supplying sC-GR migrated away from the interacting tissue-resident cells (Supplementary Fig. 8a and Supplementary Movie 3). This observation can be explained by that the relatively short half-life of reconstituted GFP on the cell membrane, which was approximately 3.5 h (Supplementary Fig. 8b). Furthermore, we confirmed that *s*GRAPHIC labeling achieved similar labeling efficiency and specificity in cell–cell interactions between human cell lines, including hematopoietic cells (Supplementary Fig. 9). Overall, the results suggested that *s*GRAPHIC is applicable for fluorescently labeling neighboring interacting partner cells across various cell types and species (Supplementary Table 1).

## *s*GRAPHIC labeling of tissue-resident cells neighboring on cancer cells in a murine metastasis model

Next, we examined *s*GRAPHIC labeling in vivo by transducing N-GR gene into liver-resident cells using adeno-associated virus serotype 8 carrying N-GR (AAV8/N-GR). The AAV8/N-GR administration transduced the N-GR gene primarily into hepatocytes, as demonstrated by the detection of mCherry in liver tissue section, flow cytometry, and scRNA-seq (Supplementary Fig. 10). Hepatocytes isolated from the liver of AAV8/N-GR-treated mice were expectedly labeled with GFP in co-culturing with cancer cells expressing sC-GR, indicating that *s*GRAPHIC labeling is functional in the primary cells (Supplementary Fig. 11a, b). Furthermore, intrasplenic injection of E0771/sC-GR cells into AAV8/N-GR-treated mice confirmed that GFP-labeled mCherry-positive cells were presented in the border area of metastatic colony–hepatic tissue (Fig. 2c). On the other hand, GFP signals were

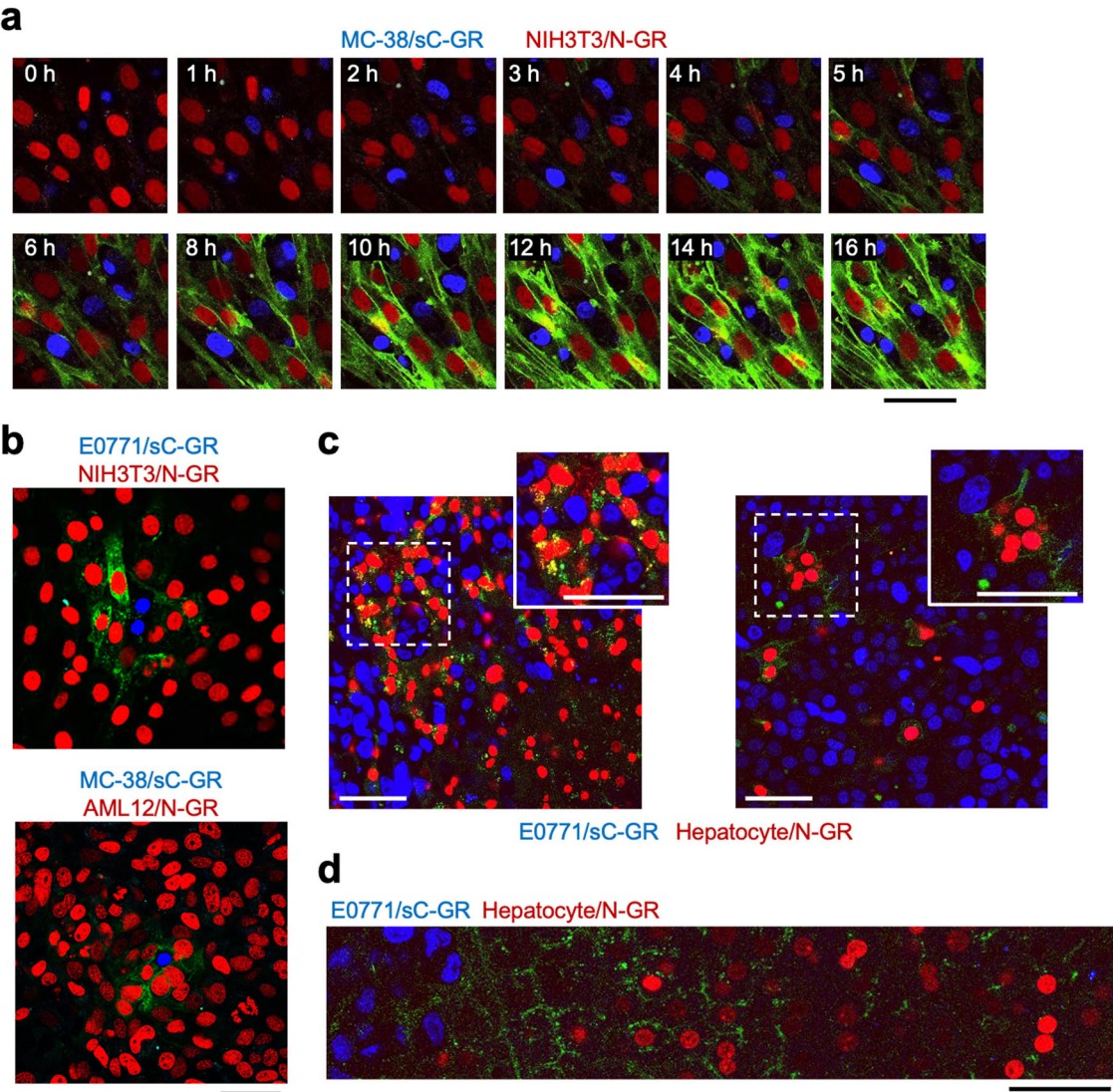

**Fig. 2 | In vitro and in vivo characterization of sGRAPHIC labeling. a** Snapshots of time-lapse fluorescence imaging for co-culturing MC-38/sC-GR cells and NIH3T3/N-GR cells from Supplementary Movie 1. Similar results were observed in independent triplicate experiments. **b** Fluorescence imaging of sGRAPHIC labeling of cancer cell−tissue-resident cell interactions (cancer cell:tissue-resident cell = 1:50). Similar results were observed in independent duplicate experiments. **c** Confocal imaging of sGRAPHIC labeling of hepatocytes neighboring on metastatic colonies in the murine liver. Confocal fluorescence imaging was performed three weeks after E0771/sC-GR cells (blue nuclei) transplantation. Similar results were observed in multiple fields of view in independent duplicate experiments. **d** Wide-field confocal imaging of sGRAPHIC labeling in the liver metastatic lesion. Confocal fluorescence imaging was performed three weeks after E0771/sC-GR cells (blue nuclei) transplantation. Similar results were observed in multiple fields of view in independent duplicate experiments. All scale bars indicate 50 μm.

not observed in metastatic colonies of E0771/sC-GR in wild-type mice and ones of E0771 in AAV8/N-GR-treated mice (Supplementary Fig. 11c). Moreover, sGRAPHIC labeling occasionally extended across several cell layers and the maximum distance of the labeling was calculated to be 99.7 ± 3.4 μm (Fig. 2d and Supplementary Fig. 11d). Collectively, we concluded that the sGRAPHIC strategy is successful in fluorescence labeling of tissue-resident cells interacting proximally with cancer cells in during metastasis.

### Characterization of metastatic niche-associated hepatocytes by sGRAPHIC with scRNA-seq

Hepatocytes are parenchymal components of the liver, and occupy 80% of the organ volume; this inevitably increases the possibility that metastasized cancer cells would neighbor on and interact with hepatocytes, but few studies have detailed functions of hepatocytes in the establishment of liver metastatic niches. According to very few studies, hepatocytes might dictate pro-metastatic niches; however, the details

are poorly understood[14,15]. Thus, to interrogate the mechanistic roles in liver metastatic niches, we combined sGRAPHIC with scRNA-seq to construct a transcriptomic platform, named Highlighting Unknown Neighbors Through Extracellular-gfp Reconstitution and Sequencing (HUNTER-seq) (Fig. 3a). We isolated and sequenced GFP+ mCherry+ (proximal) hepatocytes, GFP- mCherry+ (distal) hepatocytes from the liver with overt metastatic colonies three weeks after E0771/sC-GR cells injection (Supplementary Fig. 12a), and GFP- mCherry+ hepatocytes (control) harvested from the healthy liver after dead cell exclusion (Supplementary Fig. 12b). Differential gene expression analysis detected the upregulation of Saa1 and Lcn2 in hepatocytes (pooled proximal and distal) from the metastasized liver compared to those (control) from the healthy liver (Supplementary Fig. 13). These molecules were reported to be consistently upregulated in a previous study that described the responses of hepatocytes stimulated by soluble factors from tumor cells[16]. Gene ontology analysis of marker genes expressed in proximal hepatocytes over distal and control hepatocytes

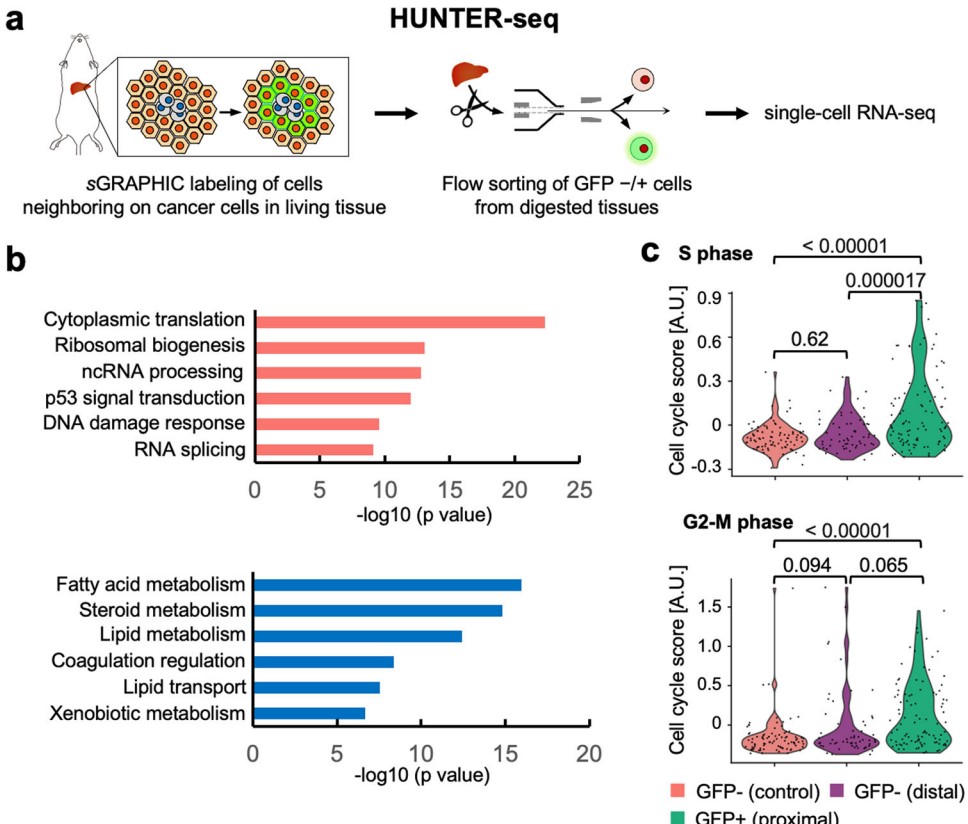

**Fig. 3 | sGRAPHIC-leveraged transcriptomics of metastatic niches in a murine model. a** Schematic of HUNTER-seq. HUNTER-seq combines sGRAPHIC, flow sorting, and scRNA-seq. **b** Gene ontology of upregulated (top) and down-regulated (bottom) genes (one-sided version of Fisher's exact test, adjusted p-value < 0.1, Log 2 FC absolute value > 0.3) in the proximal hepatocytes as compared to the distal and control hepatocytes. **c** Violin plots of cell cycling score (S phase and G2-M phase) for the hepatocytes. Cell cycling scores were plotted in arbitral unit (AU). Data were statistically analyzed with analysis of variance and Tukey honestly significant difference test (95% family-wise confidence level). The adjusted p-values are shown in the graph.

highlighted that proximal hepatocytes were characterized by stress responses and loss of liver metabolic functions (Fig. 3b). In addition, the proximal hepatocytes strongly activate ribosomal biogenesis-related signatures, which are hallmarks of cell growth and proliferation[17]. Consistently, the proximal hepatocytes exhibited significantly higher expression scores of cell-cycle-related genes than the other hepatocytes (Fig. 3c). Overall, HUNTER-seq successfully hunted metastatic niche cells and uncovered the unique gene expression signature of the proximal hepatocytes.

We further interrogated the heterogeneity of the proximal hepatocytes by exploiting the single-cell resolution of HUNTER-seq. The proximal hepatocytes were clustered into two distinct populations visualized as clusters 2 and 3 (Fig. 4a). We detected high enrichment of proximal hepatocytes in cluster 3 and termed these hepatocytes as MAHs; these cells exhibited upregulated expression of *Lgals3*, *Serpinh1*, *Ccnd2* and *Capg* (Fig. 4b). Interestingly, computational gene regulatory network reconstruction inferred that these genes were transcriptionally regulated by *Myc* in MAHs (Supplementary Fig. 14). *Myc* activation orchestrates inflammatory responses in hepatocytes;[18] this implicated that MAHs were co-opted within the inflammatory microenvironment of metastasized cancer cells as demonstrated by inflammatory gene signatures in the gene expression (Supplementary Fig. 15a, b). Among the *Myc* downstream genes expressed in the MAHs, we further focused on *Lgals3*, encoding galectin-3, which was highly expressed in GFP+ proximal hepatocytes (Fig. 4c). Notably, immuno-histochemical analysis confirmed that galectin-3 was strongly detected in the border area of the hepatic tissue and metastatic colonies but lowly or not detected in metastatic and hepatic tissues distant from the border area (Fig. 4d and Supplementary Fig. 15c). This result motivated

us to examine the impacts of galectin-3 on metastatic malignancies. EO771 cells treated with recombinant galectin-3 increased their proliferation in a dose-dependent manner (Fig. 4e). We further tested whether recombinant galectin-3 could promote the migration of cancer cells. A transwell migration assay confirmed that EO771 cells displayed enhanced migration in response to increasing concentration of galectin-3 (Fig. 4f). Consistent with these impacts of galectin-3, EO771 cells expressed one of the potent galectin-3 receptors, *Itgb1* (Supplementary Fig. 15d). Overall results implicate that galectin-3 is a pro-metastatic factor that mediates interactions between cancer cells and hepatocytes in liver metastatic lesions.

## Discussion

Here, we demonstrated that our secretory GFP reconstitution labeling system, sGRAPHIC, is a powerful tool for selectively labeling tissue-resident cells neighboring on cancer cells in co-culture systems and in murine models. This work builds on the concept established by the GFP reconstitution across synaptic partners (GRASP) technology, which was a pioneering strategy for labeling neuronal synaptic connections using split-GFP technology[19]. A main hurdle for the practical use of GRASP was the modest fluorescence from reconstituted GFP; several subsequent developments, including GRAPHIC improved the fluorescence signal of the split-GFP-based strategy to efficiently highlight neuronal interactions[20,21]. However, GRAPHIC is unable to fluorescently label cell–cell interactions involving cancer cells, probably because the highly migratory nature of cancer cells does not allow reconstituted GFPs to be maintained between cells (Supplementary Movies 2, 3). Even when cancer cells statically form cell–cell junctions, the membrane-anchored GFP fragments are not efficiently

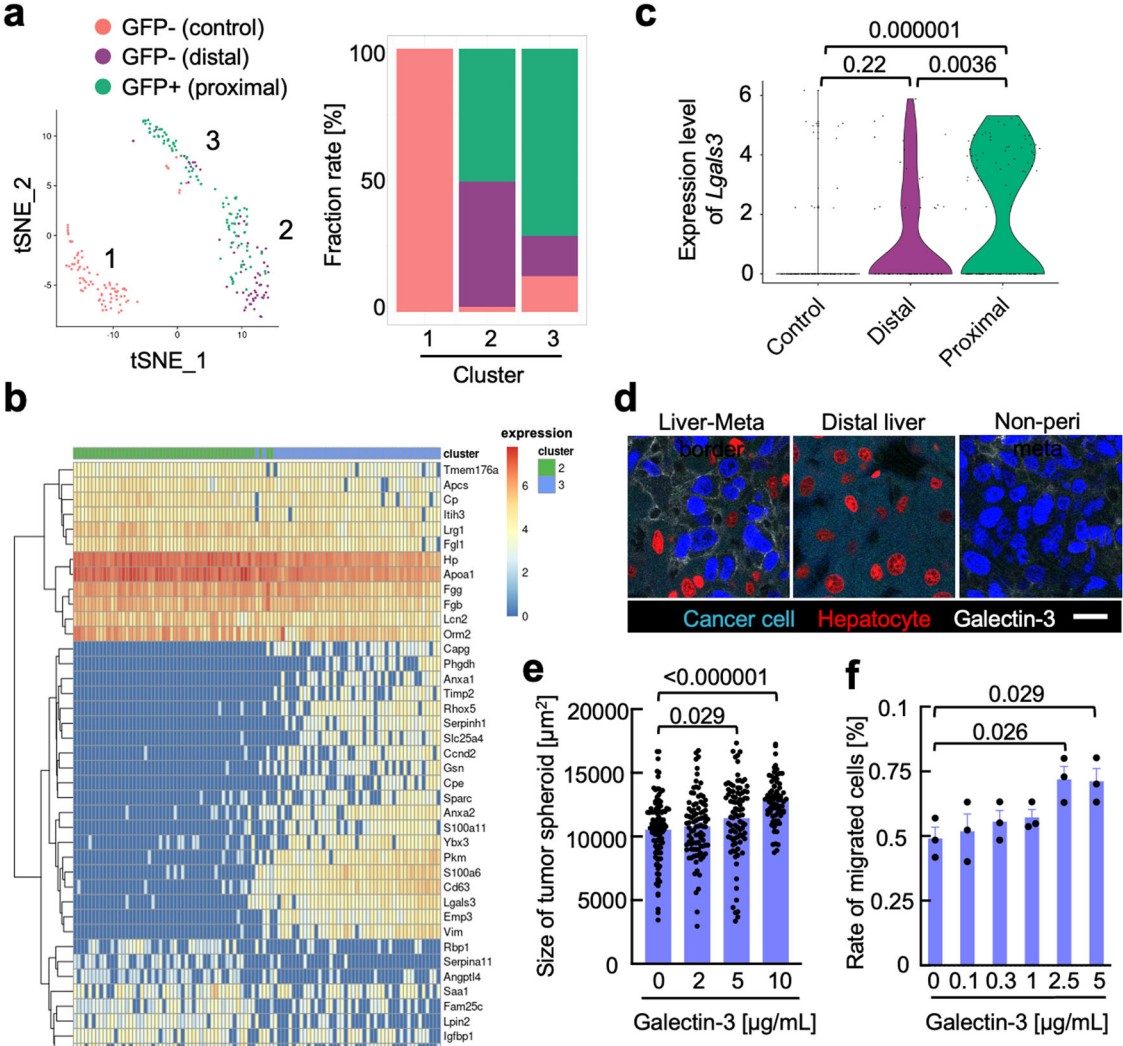

**Fig. 4 | HUNTER-seq analysis of cell–cell interactions in the liver metastatic niche. a** t-Distributed stochastic neighbor embedding (t-SNE) plot of the GFP- hepatocytes from the healthy liver (control, *n* = 87) or the metastasized liver (distal, *n* = 59) and GFP+ hepatocytes from the metastasized liver (proximal, *n* = 99). Marker genes (two-sided version of Wilcoxon Rank-Sum test, adjusted *p*-value < 0.1, Log 2 FC > 0.25) among hepatocytes (proximal, distal, control) were used for clustering. Inserted numbers indicate cluster identification (left panel). The fraction rate of the hepatocytes in each cluster identified in the t-SNE plot (right panel). **b** Heatmap displaying expression level of clusters 2 and 3 marker genes of the top 20 ranked by Log 2 FC (two-sided version of Wilcoxon Rank-Sum test, adjusted *p*-value < 0.05) expressions in the GFP+ proximal hepatocytes. **c** The expression level of *Lgals3* in the proximal, distal, and control. Data were statistically analyzed with analysis of variance and Tukey honestly significant difference test (95% family-wise confidence level). The adjusted *p*-values are shown in the graph. **d** Representative immunofluorescent staining images of galectin-3 in a

metastasized liver section. Cancer cells (H2B-Azurite), hepatocytes (H2B-mCherry), and galectin-3 signals in the area of the hepatic tissue–metastasized colonies border (Liver–Meta border), distal liver tissue (Distal liver) and non-peripheral area of the metastatic colony (Non-peri meta) are shown. Similar results were observed in independent duplicate experiments. A scale bar indicates 20 μm. **e** Tumor spheroid proliferation assay of E0771 cells treated with galectin-3 in three-dimensional culture. Data were statistically analyzed with Holm-Sidak adjusted multiple *t*-test (*n* = 102, 93, 90 and 72 for 0, 3, 5, 10 μg/mL galectin-3 respectively). The *p*-values are indicated in the graph. Data are presented as mean values ± SEM. Similar results were observed in independent duplicate experiments. Source data are provided as a Source Data file. **f** Transmigration assay of E0771 with recombinant galectin-3. Data were statistically analyzed with Holm-Sidak adjusted multiple *t*-test (*n* = 3 biologically independent samples). The *p*-values are indicated in the graph. Data are presented as mean values ± SEM. Similar results were observed in independent duplicate experiments. Source data are provided as a Source Data file.

reconstituted in an unstable intercellular space due to heterogeneous cell–cell junction proteins[22]. The distance between cell–cell membranes at intercellular junctions is highly variable because the extracellular domains of cellular junction proteins are range from several nm at the tight junction to over 100 nm at the desmosomal junction[23–25]. The polypeptide linkers used to display GFP fragments on the cell membrane are perhaps capable of bridging the distance of 30 nm, but apparently not 100 nm. Extending the extracellular linker of GRAPHIC is a potential strategy to overcome the variable distance between cells, although this strategy may not be effective in overcoming the problem of the high mobility of cancer cells. *s*GRAPHIC

successfully overcomes the shortcomings of GRAPHIC, and expands the applications of split-GFP-based labeling systems for cell–cell interactions.

Unlike neuronal physical interactions, secretory factors including cytokines and growth factors are essential for boosting cancer cell–tissue-resident cell interactions[26,27]. Paracrine interactions via these molecules can now be captured with *s*GRAPHIC. Co-culturing a small number of cancer cells with fibroblasts showed that the GFP signal was only visible within a few fibroblast layers from cancer cells (Fig. 2b). This selective labeling of the neighboring cells was made possible by the relatively short half-life of reconstituted GFP in

tissue-resident cells (Supplementary Fig. 8b). The half-life of GPI-anchored proteins is mainly modulated by the phospholipase family of proteins[28], and a recent study described that a protein domain adjacent to a GPI-attachment signal sequence defined the activity of phospholipases on GPI-anchored proteins[29]. Thus, mutated protein domains may flexibly change the half-life of reconstituted GFP on the cell membrane, providing the desired range of optical labeling from cancer cells. While, sGRAPHIC labeling in the liver metastasis tissue occasionally extended across several cell layers from cancer cells, which appeared to be longer than in the in vitro settings (Fig. 2b, d and Supplementary Fig. 11d). This may be explained by the efficient diffusion of secretory GFP fragments in narrow stromal space of living tissues. At the same time, there are several potential factors in living tissues that influence the diffusion of GFP fragments besides the Brownian motion that governs the fragment diffusion in the in vitro culture system. For example, high-density proliferation of cancer cells often increases intra-tissue pressure, resulting in a gradient of interstitial flow to surrounding healthy tissue[30]. In addition, interactions between secretory proteins and the extracellular matrix are reportedly crucial to define the diffusion of morphogens in developmental tissues[31]. Cancer cells heterogeneously produce extracellular matrices in the lesions[32]. These facts probably explain varied range of sGRAPHIC labeling in the in vivo setting (Fig. 2c, d). Therefore, we could control sGRAPHIC labeling range in living tissues through understanding the interactions between secretory GFP fragments and extracellular tissue components.

We leveraged sGRAPHIC technology to interrogate the mechanistic machinery underlying the liver metastatic niche using scRNA-seq. Because combining optical-labeling technologies with scRNA-seq is a promising approach for connecting cellular localization and functions in complex multicellular organizations, several optical-labeling techniques have recently been attempted to generate synergies with scRNA-seq. Representatively, multiple types of photoactivatable agents have enabled the isolation of cells from regions of interest in biological tissues[9,33-35]. One critical limitation of these methods is the lack of light accessibility to the deep tissues. Consequently, the ultraviolet activation of caged compounds is limitedly applied to thin dissected tissues. Even with two-photon activation of photoactivatable GFP, optical labeling is often limited to the surface layer of the organs in living animals. As we demonstrated with sGRAPHIC, fully genetically encoded systems for optical labeling do not require the step of photoactivation and offer strong benefits for labeling cells in the deep tissues of animals. In this light, the Cherry-niche system was a successful system to identify metastatic niche cells through fluorescent labeling with secretory mCherry proteins. Although the in vivo comparison between Cherry-niche and sGRAPHIC is still lacking in this study, we reproduced Cherry-niche labeling with HEK293T cells, and showed that the fluorescent labeling efficiency of sGRAPHIC surpassed that of Cherry-niche (Fig. 1c–e, and Supplementary Fig. 4). In addition, sGRAPHIC archived fluorescence labeling of multiple types of cell–cell interactions with high efficiency (Fig. 2a, b, Supplementary Fig. 5, and Supplementary Fig. 6). These results demonstrate that sGRAPHIC enables for optical labeling of transient cell–cell interactions in broad biological contexts. While, sGRAPHIC requires gene transduction of the reporters in both cancer cells and tissue-resident cells, whereas Cherry-niche requires the reporter only in cancer cells. As we employed, AAV-mediated gene transduction is a speedy strategy, but limited types of cells are targetable (Supplementary Fig. 10). To overcome this shortcoming, transgenic mice ubiquitously expressing N-GR reporter are desired for targeting diverse tissue-resident cells interacting with cancer cells.

We developed a murine liver metastasis model to characterize MAHs in the liver metastatic niche using an sGRAPHIC-leveraged scRNA-seq platform that we named HUNTER-seq. HUNTER-seq revealed that MAHs exhibited upregulated expression of the cell proliferation-related genes (Fig. 3b, c), consistent with the activation of cell-cycle progression programs in hepatocytes that occurs during the liver regeneration following liver injury[36-38]. We also detected the upregulation of Myc and its putative downstream genes in the MAHs (Fig. 4b and Supplementary Fig. 14). Suppression of Myc in hepatocytes resulted in a failure to upregulate cell-cycle control genes, thereby preventing hepatocellular proliferation[39]. Myc upregulation in hepatocytes is often observed in chronic inflammatory liver diseases including liver cancers. Consistent with these studies, MAHs were plausibly exposed to inflammation as demonstrated by the gene signature of metastasized E0771 cells and the expression of a potent inflammatory factor, Il2 (Supplementary Fig. 15a, b), which is also an inducer of Myc in hepatocytes[18,40].

Additionally, Myc may play pivotal roles in defining functional phenotypes of MAHs by regulating the expression of multiple genes, including the galectin-3 coding gene, Lgals3. Galectin-3 was reportedly secreted from hepatocytes upon liver injury[41]. Interestingly, previous studies have accumulated pieces of evidence that galectin-3 directly binds to integrins and growth factor receptors on cancer cells to activate cellular malignant processes, including cell proliferation, invasion, and chemoresistance[42,43]. Further, galectin-3 is a modulator of inflammatory cycles by directly activating various myeloid cells to boost inflammation[44,45]. Additionally, galectin-3 has been shown to activate hepatic stellate cells via cross-linking of integrin receptors, facilitating the recruitment of inflammatory bone marrow-derived myeloid cells through increased deposition of fibronectin from activated hepatic stellate cells[46,47]. Further studies are needed to detail the actions of galectin-3 on liver stromal cells in addition to cancer cells during the establishment of the inflammatory liver metastatic niche.

Tissue section-based transcriptomics have been implemented to investigate cell–cell interactions in living tissues[48-50]. This approach would provide the intact coordinates of a cell in biological tissue, but currently lacks spatial resolution or sufficient sequencing depth[51]. In addition, to identify the section containing the tiny cell population of interest, it is necessary to go through sequential sections in the organs. In this regard, sGRAPHIC allows for deep sequencing of defined single cells harvested from the whole tissues using the HUNTER-seq platform, thus addressing the shortcomings of tissue section-based transcriptomics. Furthermore, by isolating living cells of interest, the analytical capabilities of conventional live cell assays can be maximized. More importantly, connecting transcriptome with emerging single-cell multi-omics technologies, including genomics, epigenomics, proteomics, and metabolomics, would be a key to mapping a broad range of cellular statuses[52-55]. Currently, most of these multi-omics technologies are based on the dissociation of living cells from biological tissues. Therefore, sGRAPHIC can co-opt emerging multi-omics technologies, paving the way to the completion of an atlas of cell–cell interaction and bettering our understanding of tissue homeostasis and disease mechanisms.

## Methods

### Ethical statement
Ethical approval of this study protocol for recombinant DNA experiments and animal experiments was obtained from Tokyo Institute of Technology, Jichi Medical University, and RIKEN. The maximal tumor size permitted by our ethics committees or institutional review boards was 20 mm at the largest diameter in mice and was not exceeded in our experiments.

### Cell culture
The murine breast cancer cells E0771 (94A001, CH3 Biosystems, Buffalo, NY, USA) and human acute T cell leukemia cells Jurkat (TIB-152, American Type Culture Collection; ATCC, Manassas, NY, USA) were cultured in Roswell Park Memorial Institute-1640 (RPMI) medium with L-Glutamine (FUJIFILM Wako Pure Chemical Corporation, Osaka,

Japan). The murine colon adenocarcinoma cells MC-38 (ENH204-FP, Kerafast, Boston, MA, USA), murine embryo fibroblasts NIH3T3 (CRL-1658, ATCC), murine endothelial cells MS1 (CRL-2279, ATCC), human embryonic kidney cells HEK293T (632180, CloneTech, Mountain View, CA, USA), human breast cancer cells MCF7 (HTB-22, ATCC), human cervical cancer cells HeLa (CCL-2, ATCC), and pig epithelial cells LLC-PK1 (JCRB0060, JCRB Cell Bank, Osaka, Japan) and human mammary fibroblasts HMF3S (a gift of Dr. Parmjit Jat, University College London, London, United Kingdom) were cultured in Dulbecco's Modified Eagle's medium (DMEM) with high glucose (Thermo Fisher Scientific, Waltham, MA, USA). The murine osteoblasts KUSA-A1 (RCB2081, RIKEN cell bank, Saitama, Japan) were cultured in alpha Modified Eagle Minimum Essential Medium with L-Glutamine, ribonucleosides and deoxyribonucleosides medium (Nacalai Tesque, Kyoto, Japan). The murine hepatocytes AML12 (CRL-2254, ATCC) were cultured in DMEM: Nutrient Mixture F-12 medium (Thermo Fisher Scientific) with 10 μg/mL insulin (Sigma-Aldrich, St Louis, MO, USA), 5.5 μg/mL transferrin (Nacalai Tesque), 5 ng/mL selenium (Sigma-Aldrich) and 40 ng/mL dexamethasone (Sigma-Aldrich). All media contained 10% fetal bovine serum (FBS) (Thermo Fisher Scientific), 100 IU/mL penicillin/streptomycin (Nacalai Tesque) and were used for culturing cells in a humidified incubator at 37 °C, with 5% $CO_2$. All cell lines were regularly tested for mycoplasma contamination and were authenticated by morphology check and growth curve analysis.

## Plasmid construction
n-GRAPHIC and c-GRAPHIC in CSII-CMV vectors were gifted from Dr. A. Miyawaki (RIKEN, Saitama, Japan)[11]. To construct plasmids for encoding secretory GRAPHIC reporters (secretory n-GRAPHIC and secretory c-GRAPHIC in CSII-CMV vectors), we deleted the GPI-anchored domain from n-GRAPHIC and c-GRAPHIC by using In-Fusion HD Cloning Kit (Takara Bio, Shiga, Japan). For the HiBiT assay of GFP fragment secretion, synthetic cDNAs encoding ss-sfGFP-NT-LZA-HiBiT or ss-sfGFP-CT-LZB-HiBiT were obtained (Eurofins Genomics, Tokyo, Japan), and then flanked with T2A-H2B-mCherry or T2A-H2B-Azurite in CSII-CMV-MCS lentiviral backbone by using In-Fusion HD Cloning Kit (Takara Bio).

Synthetic Cherry-niche cDNA was obtained (Eurofins Genomics), and inserted into the CSII-CMV-MCS lentiviral backbone (RIKEN BRC, Saitama, Japan). To assess the labeling efficiency of Cherry-niche labeling, H2B-Azurite in CSII-CMV vector was constructed by inserting the amplified sequence of H2B-Azurite into the CSII-CMV-MCS lentiviral backbone.

## Stable gene transduction with lentivirus vectors
For production of lentivirus particles, the CSII plasmids were co-transfected with the packaging plasmid psPAX2 (#12260, addgene, Watertown, MA, USA), the VSV-G- and Rev-expressing plasmids (pCMV-VSV-G-RSV-Rev) (RIKEN BRC) into HEK293T cells (CloneTech) by PEI MAX (Polysciences, Warrington, PA, USA). After 48 h, the supernatants were harvested to concentrate the lentivirus particles by centrifugation with Lenti-X™ Concentrator (CloneTech). All cells, HEK293T, HeLa, E0771, MC-38, MCF7, NIH3T3, MS1, KUSA-A1, AML12, Jurkat, and HMF3S, were cultured for 48 h in the medium with the lentivirus particles and 10 μg/μL polybrene (Sigma-Aldrich). The successfully transduced cells were selected by fluorescence-activated cell sorting (SH800, SONY, Tokyo, Japan) for expression of mCherry or Azurite.

## sGRAPHIC labeling in vitro
The cell lines with constitutive expression of sGRAPHIC reporter genes (sC-GR and N-GR) ($3.0 \times 10^5$ cells each) were co-cultured for 24 h in 35-mm glass-bottomed dishes (Eppendorf, Hamburg, Germany) or 6 well plates. The co-cultured cells were then observed with confocal fluorescent microscopes (Zeiss LSM 780, Carl Zeiss, Oberkochen,

Germany or Leica TCS SP8, Leica Microsystems, Wetzlar, Germany) or analyzed with flow cytometer SH800 (SONY). For time-lapse imaging, fluorescence images were acquired at every 10 min in a stage-top $CO_2$ incubator (Tokai Hit, Shizuoka, Japan). The fluorescent images were displayed by maximum intensity projection with image stacks focusing on both cancer cells and stromal cells. Maximum intensity projection was constructed with NIH ImageJ/Fiji open-source software[56]. For flow cytometry analysis, the cells were harvested with Cell Dissociation Solution (Biological Industries, Cromwell, CT, USA) and were resuspended into phosphate-buffered saline (PBS) containing 2% FBS. To study localization of reconstituted GFP, co-cultured cells were treated with Cell Dissociation Buffer (Biological Industries), 0.25% Trypsin-EDTA, 0.5% Collagenase (FUJIFILM Wako Pure Chemical Corporation) or 0.5% Collagenase plus 0.1% trypsin inhibitor (FUJIFILM Wako Pure Chemical Corporation). Flow cytometry data were analyzed with FlowJo™ 10 (Becton, Dickinson and Company, Sparks, MD, USA).

For sGRAPHIC labeling with primary hepatocytes, hepatocytes were isolated from AAV8/N-GR-treated murine livers. The hepatocytes ($4.0 \times 10^5$ cells) were seeded on 35-mm glass-bottomed dishes (Eppendorf) or 6 well plates for 3 h in DMEM containing 10% FBS. After 3 h, the medium was replaced with William's E Medium (Thermo Fisher Scientific) containing 1% glutamine (Biological Industries) and 10% FBS (Thermo Fisher Scientific), and started co-culturing with E0771/sC-GR cells or MC-38/sC-GR cells ($4.0 \times 10^5$ cells). After 24 h of co-culturing, the cells were observed with a confocal microscope Olympus-FV3000 (Olympus, Tokyo, Japan) and were analyzed with flow cytometry BD FACSAria™ (Becton, Dickinson and Company).

## Cherry-niche labeling in vitro
HEK293T/Cherry-niche cells and HEK293T/H2B-Azurite cells ($3.0 \times 10^5$ cells each) were co-cultured for 24 h. The co-cultured cells were observed with a confocal fluorescent microscope Leica TCS SP8 (Leica Microsystems) and were analyzed with flow cytometer SH800 (SONY).

## Mice for in vivo experiments
Female C57BL/6 albino mice were obtained from Charles River Laboratory Japan (Yokohama, Japan). All mice used were age-matched (5 weeks of age) females, were provided access to food and water *ad libitum*, and were housed in the animal facilities at Tokyo Institute of Technology, RIKEN or Jichi Meidcal University. The animal facilities were maintained at 20–25 °C with 40–60% humidity under a standard 12-h light–dark cycle. the experimental procedures using mice were approved by the Animal Experiment Committees of Tokyo Institute of Technology (authorization number 2019-031), RIKEN (authorization number W2020-191770) and Jichi Medical University (authorization number 21003), and carried out in accordance with relevant national and international guidelines.

## Isolation of primary hepatocytes from the murine liver
Primary hepatocytes were isolated by a two-step collagenase perfusion procedure[57]. The reagents were summarized in Supplementary Table 2. After mice were anesthetized with three types of mixed anesthetic agents (0.3 mg/kg of medetomidine (Meiji Seika Pharma, Tokyo, Japan); 4.0 mg/kg of midazolam (Maruishi Pharma, Osaka, Japan); 5.0 mg/kg of butorphanol (Meiji Seika Pharma)), the liver was perfused with the pre-perfusion solution followed by collagenase-containing solution from the inferior vena cava into the portal vein. After 10 min of liver perfusion, the liver was dissected and mechanically destructed to harvest liver cells in DMEM containing 10% FBS. The suspension of liver cells was forced through a 100 μm Cell Strainer (Greiner Bio-One, Kremsmünster, Austria). The filtered suspension was centrifuged at $50 \times g$ for 5 min at 4 °C and resuspended in DMEM containing 10% FBS. This centrifugation and resuspension were repeated three times.

### Gene transduction with adeno-associated virus vectors

Virus particles of AAV serotype 8 carrying N-GR (AAV8/N-GR) were provided by Vector Builder (Silicon Valley, CA, USA). C57B/6 albino mice (female, 6 weeks old) were intravenously administrated with AAV8/N-GR ($5.0 \times 10^{12}$ GC/body) suspended in PBS.

### A murine model of liver metastasis

C57B/6 albino mice (female, 7–8 weeks old, Charles River Laboratory Japan) were anesthetized with isoflurane (FUJIFILM Wako Pure Chemical Corporation). E0771/sC-GR cells ($1 \times 10^6$ cells) suspended in 100 μL PBS were injected into the spleen of anesthetized mice using 29 G syringe needle in 30 s.

### sGRAPHIC labeling in vivo

One–two weeks after AAV8/N-GR transductions into murine liver tissues, E0771/sC-GR cells were transplanted into the spleen. Three weeks after cancer cells transplantation, murine livers were dissected and then observed with Leica TCS SP8 confocal microscope (Leica Microsystems). The fluorescence images were analyzed with NIH ImageJ/Fiji open-source software[56].

### Cell isolation for HUNTER-seq

E0771/sC-GR cells ($1.0 \times 10^6$ cells) were injected into the spleen a week after AAV8/N-GR treatment of mice. Three weeks after cancer cells transplantation, primary hepatocytes and cancer cells were harvested from dissociated murine livers through the two-step collagenase perfusion procedure. In this step, we selectively destructed an area with a visible metastatic colony to increase yields of GFP-positive hepatocytes. The harvested cells were fluorescently sorted into 96-well plates for scRNA-seq with BD FACSAria ™ (Becton, Dickinson and Company). Dead cells were excluded with Fixable Viability Stain 700 (BD Biosciences, San Jose, CA, USA) staining. Doublet cells were eliminated by SSC values. We gated mCherry and GFP-positive cells for the proximal hepatocytes, mCherry-positive and GFP-negative cells for the distal or control hepatocytes, and Azurite-positive cells for metastasized E0771/sC-GR cells.

### Library preparation for HUNTER-seq

Cells were sorted into 96-well plates containing 4.25 μL of lysis solution (Takara Bio), including barcoded oligo-dT28 primers, flanking 8-bp UMI, 10-bp cell barcode, TruSeq read 1 sequence, and PCR handle (12 pmol) and 1 U RNase Inhibitor (Takara Bio). During single-cell sorting, 96-well plates were kept at 4 °C. Immediately after cell sorting, each plate was temporarily sealed with MicroAmp Optical Adhesive Film (Thermo Fisher Scientific) and spun to ensure cell immersion into the lysis solution.

Reverse transcription was carried out by incubating reaction mixtures at 42 °C for 90 min and 70 °C for 10 min in a 10 μL of volume containing 10 U/μL of reverse transcription enzyme, 1× First Strand Buffer, 2 mM of dithiothreitol (SMARTScribe, Takara Bio), 1 U/μL RNase Inhibitor (Takara Bio), 1 mM dNTP (Thermo Fisher Scientific) and 2.4 μM template switching oligos (QIAGEN, Hilden, Germany). The excess reverse transcription primers were digested using 2.5 U Exonuclease I (Takara Bio) at 37 °C for 30 min and 80 °C for 20 min. The synthesized cDNA was denatured at 95 °C for 1 min followed by 15 cycles of PCR using SeqAmp DNA polymerase (Takara Bio) (98 °C for 10 s, 65 °C for 30 s, 68 °C for 4 min), and followed by 72 °C for 10 min. The amplicons of 16 single cells were pooled and purified with 0.7× AMPure XP (Beckman Coulter, Brea, CA, USA). Each pooled sample was assessed the quality with Agilent 2100 Bioanalyzer (Agilent Technologies, Santa Clara, CA, USA) and Qubit 4.0 Fluorometer (Thermo Fisher Scientific), as well as quantitative PCR targeting *Gapdh* (glyceraldehyde-3-phosphate dehydrogenase, Mm99999915_g1 (mouse Gapdh); Thermo Fisher Scientific). The pooled samples (400 pg per sample) were indexed with Nextera XT DNA Library Preparation Kit

(Illumina, San Diego, CA, USA) with TruSeq i5 primers and Nextera i7 primers and purified with 0.6× AMPure XP beads (Beckman Coulter). The yielded libraries were sequenced with 150 bp pair-end read using HiseqX platform (Illumina).

### Data analysis for HUNTER-seq

The raw reads were preprocessed with UMI-tools (1.0.0)[58] and demultiplexed by *fqtools* (2.1)[59] and *subseq* of seqtk (1.3-r106) (https://github.com/lh3/seqtk). Input reads were downsampled to be 23,918 reads per cell with seqtk. Mapping of sequence reads to a reference genome (GRCm38) was done with STAR (2.7.9a)[60]. The aligned reads were annotated by *featureCounts* (2.0.1)[61] with Mus_musculus.GRCm38.102.gtf. The cell barcodes and unique molecular identifier (UMI) were quantified with UMI-tools (1.0.0)[58]. Single-cell sequence data were analyzed using the Seurat R package (4.3.0)[62,63]. We filtered out cells with more than 7% mitochondrial gene expression or fewer than 3000 unique transcripts from the analysis. The counts were normalized with a *LogNormalize* and scaled with a scale factor of 100,000. The *VlnPlot* function in Seurat was used for visualizing expression levels of genes of interest. Differentially expressed genes (DEGs) among two groups were identified using the *FindMarkers* function and the Wilcoxon Rank-Sum test in Seurat by thresholding at defined adjusted *p*-value and Log 2 fold change (FC) (Fig. 3b, Supplementary Fig. 13, and Supplementary Fig. 15a). The marker genes of clustered populations were identified by comparing the cells in a population with all other cells with the *FindAllMarkers* function with the Wilcoxon Rank Sun test and only.pos = T option in Seurat (Fig. 4a, b, and Supplementary Fig. 14). Functions enriched with the up- or down-regulated genes (DEGs with positive and negative Log2FC) were respectively analyzed by the GO functional annotation in clusterProfiler (4.6.0)[64] on org.Mm.eg.db (3.16.0)[65] (Fig. 3b). S and G2-M phase scores were calculated using the *CellCycleScoring* in Seurat with mouse cell-cycle phase genes (https://github.com/hbc/tinyatlas/blob/master/cell_cycle/Mus_musculus.csv). S and G2-M phase scores were statistically compared among proximal, distal and control hepatocytes with analysis of variance and Tukey honestly significant difference test and visualized by the *VlnPlot* function in Seurat (Fig. 3c). The PCA on the single-cell expression matrix was performed with the *RunPCA* function and the marker genes for hepatocyte groups (proximal, distal, and control). Clustering of hepatocytes was performed the *FindNeighbors* and the *FindClusters* function. t-Distributed stochastic neighbor embedding (t-SNE)[66] was used for visualization (Fig. 4a). The proximal and distal hepatocytes were ordered in Fig. 4b by the *DiffusionMap* function in *destiny* (3.12.0)[67] with the top 20 upregulated genes in clusters 2 and 3, and the levels of gene expressions were visualized by using CRAN *pheatmap* (1.0.12) (Fig. 4b). Gene regulatory networks of cluster 3 were analyzed by SCENIC (1.2.4)[68,69] (Supplementary Fig. 14).

### Histological analysis

The isolated liver was immediately frozen in optimal cutting temperature compound (Sakura Finetek Japan, Tokyo, Japan), and stored at −80 °C. The frozen liver was sliced into 10 μm sections using Leica CM3050S cryostat (Leica Biosystems, Wetzlar, Germany), and fixed in 4% paraformaldehyde phosphate buffer solution (Nacalai Tesque) for 10 min at room temperature. For hematoxylin-eosin staining, the liver sections were stained with hematoxylin and eosin (FUJIFILM Wako Pure Chemical Corporation) for 3 min and 2 min, respectively. For immunofluorescence staining, the fixed sections were washed with PBS containing 0.05% Tween-20 (Nacalai Tesque) (PBS-T) three times and then were incubated in blocking buffer (5% donkey serum, FUJIFILM Wako Pure Chemical Corporation) for 1 h at room temperature, and then incubated with anti-mouse galectin-3 primary antibody (CL8942AP, 1:2000, Cedarlane Laboratories, Burlington, Canada) diluted in PBS containing 1% bovine serum albumin (Nacalai Tesque)

overnight at 4 °C. After washing with PBS-T, the samples were incubated for 45 min at room temperature in the dark with secondary antibodies conjugated with Alexa Fluor 633 (A21094, 1:1500, Thermo Fisher Scientific) in PBS containing 0.2% bovine serum albumin (Nacalai Tesque). The sections were washed with PBS-T three times, and then mounted on slides with a mounting medium (Vector Laboratories, Burlingame, CA, USA). Fluorescent images were acquired with a Leica TCS SP8 confocal microscope (Leica Microsystems). The fluorescence images were analyzed with NIH ImageJ/Fiji open-source software.

## Tumor spheroid proliferation assay

E0771/mCherry-luc2 cells ($2 \times 10^4$ cells / well) were seeded in 96-well plates optimized for tumor spheroid formation (IWAKI EZSPHERE, AGC Techno Glass Co. Ltd., Shizuoka, Japan) with 10% FBS-RPMI with murine galectin-3 (Biolegend, San Diego, CA, USA), and incubated for 72 h. The mCherry fluorescence was detected to visualize tumor spheroids (Keyence, Osaka, Japan). The size of tumor spheroids was analyzed in three independent images for each concentration of galectin-3 with Hybrid Cell Count software (Keyence).

## Transmigration assay

E0771/mCherry-luc2 cells ($2 \times 10^4$ cells / well) were labeled with 2 μmol/L CellTracker® Green (Thermo Fisher Scientific) for 30 min. After washing with PBS, the cells were seeded in the top filters with 8-μm-pore FluoroBlok (Corning, Corning, NY, USA), and then the top filters were placed on 24-well containing RPMI medium supplemented with 10% FBS and varied concentrations of murine galectin-3 (Biolegend). After a 24-h incubation, the migrated cells on the bottom side of the top filter were directly observed with a fluorescence microscope (Keyence). The migrated cells were analyzed with Hybrid Cell Count software (Keyence). Results were presented as a rate of the occupied area by migrated cells in the field of view.

## Statistics and reproducibility

Data are presented as means ± standard error of the mean (SEM) and were statistically analyzed with two-tailed Student's t-test or multiple *t*-test. *P*-values < 0.05 were considered statistically significant. Statistical analyses and graphs were performed using GraphPad Prism 9 (GraphPad Software, San Diego, CA, USA).

## Reporting summary

Further information on research design is available in the Nature Portfolio Reporting Summary linked to this article.

## Data availability

The scRNA-seq data generated in this study have been deposited in the NCBI BioProject under accession code PRJNA841462. Due to size limitations, the raw microscopy data can be made available within 2 weeks upon request to T. Kuchimaru. Source data are provided with this paper.

## Code availability

The algorithm that supports the findings of this study is available on GitHub: https://github.com/Minegishi-Misa/sGRAPHIC-HUNTER.git[70].

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

## Acknowledgements

We thank K. Aoki (National Institute of Basic Biology, Japan) for the construction of lentivirus vectors in the establishment of cell lines expressing *sGRAPHIC* reporters. We thank N. Kinoshita (RIKEN, Japan) for insightful discussion on *sGRAPHIC* design. We thank Y. Suzuki, A. Hirasawa, H. Miyauchi, K. Ishikawa (Jichi Medical University, Japan), and R. Takahashi (RIKEN, Japan) for technical assistance in cell culture and animal experiments. We thank K. Ohtawa and K. Fukumoto (Support Unit for Bio-Material Analysis, RRD, CBS, RIKEN, Japan) for technical assistance with cell sorting and scRNA-seq. We thank Y. Hayakawa (Center for Cytometry Research, Jichi Medical University, Japan) for technical assistance with cell sorting. We thank G. Kondoh (Kyoto University, Japan) for kindly providing cDNA of mouse preproacrosin signal peptide and mouse Thy-1 GPI-anchored domain, the RIKEN CBS-Olympus Collaboration Center (RIKEN, Japan) and the Open Research Facilities for Life Science and Technology (Tokyo Institute of Technology, Japan) for

fluorescence microscopy. We thank P. Jat (University College London, United Kingdom) for kindly providing HMF3S cells. This research was supported by the Japan Society for Promotion of Science (KAKENHI grant no. JP17H04989, JP19H04814, JP20H02862 to T.Ku., JP21K18194 to H.S. and T.Ku., JP20J15062, JP22J00672 to M.M.), the CREST program of the Japan Agency for Medical Research and Development (grant no. JPMJCR2124 to H.S. and T.Ku.), the PRIME program of the Japan Agency for Medical Research and Development (grant no. JP20gm6210028 to N.T.), the Vehicle Racing Commemorative Foundation (to T.Ku, T.I. and N.T.), Takeda Science Foundation (to T.Ku.), and Yoshida Scholarship Foundation (to M.M.).

## Author contributions

T.Ku. conceptualized the study. M.M., T.Ku., T.Shi. and A.M. designed sGRAPHIC. M.M., T.Ku., H. Hamana and H.K. established sGRAPHIC cell lines. M.M. and T.Ku. performed fluorescence imaging. M.M, T.Ku. and T.I. performed in vitro cellular assays. M.M., T.Ku. and H. Hara performed flow cytometry. M.M., T.Ku., S.I., S.M., M.S., S.W. and A.Su. isolated primary hepatocytes. M.M. and T.Ku. performed histological analysis. M.M., K.N., K.I., A.Shi. and H.S. performed HUNTER-seq. M.M., T.Ku., K.N., T.I., S.I., K.I., S.M., M.S., S.W., A.Shi., Y.M., T.Sat., D.S., S.S., Y.H., A.Su., T.Ko., T.Ka., A.M., N.T., H.S, S.K.-K. and S.N. discussed data. M.M., T.Ku., S.S., T.Shi., N.T., H.S., S.K.-K. and S.N. wrote the manuscript. T.Ku. and H.S. supervised the project.

## Competing interests

The authors declare no competing interests.

## Additional information

[1]School of Life Science and Technology, Tokyo Institute of Technology, Kanagawa, Japan. [2]RIKEN Cluster for Pioneering Research, Saitama, Japan. [3]Graduate School of Medicine, Jichi Medical University, Tochigi, Japan. [4]Center for Molecular Medicine, Jichi Medical University, Tochigi, Japan. [5]Data Science Center, Jichi Medical University, Tochigi, Japan. [6]RIKEN Center for Brain Science, Saitama, Japan. [7]Institute for Tenure Track Promotion, University of Miyazaki, Miyazaki, Japan. [8]Faculty of Science and Engineering, Kindai University, Osaka, Japan. [9]Medical Institute of Bioregulation, Kyushu University, Fukuoka, Japan. [10]MediGear International Corporation, Kanagawa, Japan. [11]Graduate School of Medicine, Juntendo University, Tokyo, Japan. [12]School of Medicine, Fujita Health University, Aichi, Japan. [13]Department of Immunology, Faculty of Medicine, Academic Assembly, University of Toyama, Toyama, Japan. [14]Clinical Pharmacology, Jichi Medical University, Tochigi, Japan. [15]These authors jointly supervised this work: Takahiro Kuchimaru, Hirofumi Shintaku. ✉e-mail: kuchimaru@jichi.ac.jp; hirofumi.shintaku@riken.jp

