## [Peer Review File · Nature Communications]

Secretory GPF reconstitution labeling of neighboring cells interrogates cell–cell interactions in metastatic nichesREVIEWER COMMENTS

Reviewer #1 (Remarks to the Author):

Minegishi et al., have developed a method to fluorescently label in proximal cells in tissue by modifying a technology previous developed in a different lab (Kinoshita et al., 2019 and 2020). Indeed, the corresponding authors of the previous paper are also contributing authors here.

The approach is based on split GFP approach largely used to study protein-protein interaction (review in Romei and Boxer, 2019 10.1146/annurev-biophys-051013-022846).

In Kinoshita et al., 2019 split GFP technology was used by fusing the N-terminal or the C terminal to glycosylphosphatidylinositol anchor domain and express in two different cells (GRAPHIC). Here GFP signal would reconstitute only if the two expressing cells are in contact with each other. In the present work Minegishi et al., modified this approach to make one of the GFP fragment soluble and not anchored to the plasma membrane allowing for the producing cells to trigger a membrane GFP signal in neighbouring cells expressing the membrane anchored fragment. The author presents a proof of concept for an in vivo application of this approach in the context of hepatocyte-cancer cells interaction during liver metastasis. In general, the authors present a very neat system, that could lead to the implementation of new genetic mouse models to fluorescently detect cells proximal to specific cells in a tissue. I feel that the manuscript's main value is presentation of this model system with a proof-of-concept validation of its in vivo application. In my opinion, the work is in principle suitable for publication in Nature Communication. However, the current version suffers for some data overinterpretation, more details need to be provided for the in vivo validation and some part of the text need to be revised.

Major comments on state-of-the-art contextualization and objectivity

1. There are certainly many discoveries to be made in the cancer field and so much we still do not know, but the notion of the "metastatic niche" formation it is not one of them. When typing "metastatic niche" in pubmed, there are currently 1,854 studies only in the last 5 years. While it is a fact that how such complex phenomenon is governed or tailored to different cancer cells it is subject of intense investigation, I found rather unacceptable how the authors start the manuscript (abstract and introduction) in a way that put in doubt the very existence of such well known phenomenon. The authors use sentences like "hypothetically establishing a metastatic niche", "metastatic niche hypothesis", "...cancer cells hypothetically form metastatic niche".

This reviewer agrees that a great limitation in studying the metastatic niche are tools that allow to detect tissue cells proximal to metastatic cancer cells, but we certainly do not need those tools to know that the metastatic niche exist.

The manuscript should be written in a way to objectively describing the advantages of the sGRAPHIC in the context of the knowledge of the field.

2. On the similar line, when presenting the split GFP strategy both advantages and limitation need to be clearly described in order to avoid too biased messages. For examples, when comparing the efficiency of sGRAPHIC with the Cherry-niche system the authors clearly show that the reconstituted GFP signal is

much brighter compared to the internalized mCherry signal in recipient cells, and that in vitro, sGRAPHIC can label a large proportion of cells when expressed in this particular cell line. In the discussion session, the authors do contextualize the result to the cell type used, but they also need to add that conclusion on the comparison of the two systems cannot be fully made, as they were not compared in the most relevant in vivo approach. Moreover, the authors also need to discuss that the main difference in the two systems: namely that in the sGRAPHIC both recipient and donor cells need to be genetically modified. Therefore, to achieve an unbiased detection of the various cells surrounding cancer cells in vivo, the sGRAPHIC will require the generation of a N-GR transgenic mouse model under a ubiquitous promoter, like Actin promoter. I believe that generating the mice would not be required for publication, but it should be at least mentioned in the discussion.

Major comments on presented data

The in vitro experiments describing the sGRAPHIC system are overall well-presented and described. The main concerns are around the in vivo validation.

1. Given the fact that by using the adenovirus delivery of the N-GR construct, the data show that cells targeted are hepatocytes (Fig S10), the fact that in this proof of concept that sGRAPHIC can detect tissue cells surrounding cancer cells in the tissue is limited to hepatocytes.

2. Given the strong GFP signal shown in the co-culture as well as the bright GFP signal in the liver metastasis in Fig 2c, can the author explain why, in the FACS plot of figure S12 the population of GFP-labelled hepatocytes is hardly visible? Can the authors provide a quantification of the ratio of Azurite+ cancer cells vs GFP+ hepatocyte in liver metastasis?

Additionally, when comparing the GFP signal of GFP- hepatocytes to the one of healthy hepatocytes, it looks like there is a shift up in the overall population (Fig S12), can the author provide an explanation for this? Can the author estimate the distance of GFP labelling from the producing cancer cells?

3. In the in vivo setting the healthy liver refers to mice that did not receive cancer cells, but where the mice injected with the adenovirus to label the hepatocytes? Of course, this would have been the correct control to use and based on Fig S12, as the hepatocytes express Cherry, I assume this was the case. However, Fig 4a shows that the main difference in hepatocyte signature is the presence of cancer cells in the liver, compare to that, the difference between hepatocytes near or far from the cancer cells is much less evident. Is this possible? In absence of primary tumour, when injecting cancer cells directly into the tissue, why would hepatocytes far from seeding cancer cells be so deeply influenced? At an early stage of metastatic colonization, this is very unlikely. To induce such dramatic whole tissue effect the metastatic load would need to be very high, how big were the metastases when the livers were analysed? Can the authors provide an explanation for this data?

4. The authors provide a very general analysis of the single cell sequencing data, which is fine as a proof of concept, but they also need to acknowledge that mechanistic conclusions cannot be made without a functional validation. The message that the expression of Galectin-3 found in hepatocytes in the niche can support cancer cell growth is supported by a functional validation at least in vitro. But there is no data showing that Galectin-3 in hepatocytes is important for liver metastasis in vivo, therefore, the conclusion needs to be toned down. To determine Galectin-3 requirement in vivo, the authors could use Adenovirus to deliver shRNA to knock down Galectin-3 in the hepatocytes to show that in vivo this leads to reduced metastatic efficiency.

5. No conclusions should be made on functional value of Myc activation in hepatocyte. Unless a similar KD Myc in hepatocyte approach using adenoviruses suggested above is made to show its requirement in vivo. Without this functional evidence, the Myc activation should not be presented in the model in Fig S16 and summary image.

6. Similarly, no conclusions should be made on the induction of Myc via IL2, the fact that it is found expressed in cancer in the liver certainly does not provide a causative link for Myc activation in the hepatocyte. This data can be maintained in the discussion as a speculative possibility but must be removed from the model in Fig S16 and summary image.

I personally do not think that the authors need to provide necessarily a novel mechanistic inside a about liver metastasis, but the data of the in vivo labelling need to be clearer, with more imaging of labelling, distance from the cancer cells, metastatic size and cancer cell to labelled hepatocyte ration.

Reviewer #2 (Remarks to the Author):

Key results

The authors of the manuscript built upon an existing and previously published optical labelling system GRAPHIC to make an improved version of it called sGRAPHIC to specifically study cell-cell interactions in metastatic niches. They combined sGRAPHIC with single-cell RNA sequencing thus creating the HUNTER-seq platform. This platform was used to study the gene expression patterns in metastatic niche-associated hepatocytes (MAHs) in a murine model of liver metastasis. Galectin-3 expression was found to be upregulated in those MAHs. Galectin-3 is known as a pro-metastatic factor.

Validity

I would like to see a clearer description of differences of sGRAPHIC from a previously published GRAPHIC system. In what way is the system new and improved? Second, was power analysis used as part of statistical analysis to estimate the necessary sample size(s) to look for significant differences?

Significance

The manuscript describes in great detail a new and very useful application of the previously published optical labeling system combined with additional types of analysis, like flow cytometry and scRNA-seq, to study cell-cell interactions more effectively in metastatic niches, combined with the study of gene expression changes. Use of cells of different cancer types allows for wider application of the proposed sGRAPHIC system.

The previously published system is described in Kinoshita, N. et al. Genetically Encoded Fluorescent Indicator GRAPHIC Delineates Intercellular Connections. *iScience* 15, 28-38 (2019).

<https://doi.org:10.1016/j.isci.2019.04.013>

Data and methodology

Overall, in my assessment, the used approach is valid, the quality of all the data including the

supplemental materials is sufficient, and the quality of presentation can be improved in terms of clarity.

Analytical approach

Since I am not an expert in optical labeling systems, It is hard for me to evaluate the validity of analytical approach in this regard. I would like to know if power analysis was used as part of statistical analysis to estimate the necessary sample size(s) to look for significant differences?

Suggested improvements

I have included a number of suggested improvements in my comments added to the pdf of the manuscript (uploaded). I think the manuscript present a solid piece of existing systems and technology application with valid and useful results, but improvements are needed in manuscript clarity. I highly recommend adding a list of used abbreviations.

Clarity and context

In my view, the clarity of the text can be improved, with adding the explanations why the presented types of analysis were used and what is innovative or novel in the used approaches. I agree that the results have been provided with sufficient context and consideration of previous work.

References

The manuscript references previous literature appropriately.

My expertise

I don't have expertise in constructing, using or evaluating optical labeling systems.

Reviewer #3 (Remarks to the Author):

This paper is to develop the sGRAPHIC system as a new genetic optical labeling system, based on previous GRAPHIC system by adding secretary-C terminal GR (sC-GR, I can't find what's GR for in this paper) that can bind the N- terminal GR (N-GR) to activate green florescence and use it to understand the cell-cell interaction (CCI) between metastasized cancer cells and different tissue-resident cells. Several aspects of the development of this technique have been evaluated and optimized, including the pair of sC-GR and N-GR, the half-life of reconstituted GFP, labelling performance both in vitro and in vivo. Particularly, HUNTER-seq was further developed as application of sGRAPHIC combined with the single cell RNAseq (scRNAseq) technology to sequence the obtained fluorescent neighbor tissue-resident cells near to metastasized liver tumor cells (from breast cancer cell line originally). Candidate genes and pathways have been identified with promising interpretations based on previous literature. This is a very intriguing novel procedure to study an important question of CCI between metastasized cancer cells and different tissue-resident cells that may provide microenvironment of metastasis. The paper also described and discussed the limitation of previous methods. Overall, the work sup-ports the conclusion.

The paper is carefully well-written. However, there are spaces to improve this work in terms of the structure and the content for clarification of the developed method and reproducibility.

Major comments (mostly in data analysis):

1, In Line 193-194, three groups of hepatocytes (1) proximal, (2) distal and (3) from healthy controls. In Line 195 “Differential gene (DE) expression analysis”, it’s not clear whether DE genes were obtained by comparing the pooled proximal and distal hepatocytes (from metastasized samples) to the hepatocytes in the control group. If so, the later methods or this section need to clarify it. In Methods, “analysis of variance” in Line 670 and “Rank Test” in Line 681 were mentioned, but neither seems exactly reflects the DE analysis here.

Same question about sample group comparison remains in Line 199-201 for gene ontology analysis. Is it two group comparison or paired of three group comparisons? What (logFC? proximal vs each of the other two or pooled of the other two?) are based to rank genes?

2, The caption in Figure S14 needs more details. What comparisons are the “up-regulated gene” from? How many of them? Are all of them connected as in S14? How correlation was calculated in SCENIC and what’s the cutoff of correlation for edges in the network? Some related details are in Methods, but not quite sufficient to allow clear understanding.

Line 705-706, “Highlighted gene regulatory
706 networks in cluster 3 with CoExWeight more than 0.007”. It’s not clear what does this mean. Overall, the procedure of network analysis needs to be further clarified.

3, The Methods section is a bit difficult to follow. It’s not always easy to tell which method procedure corresponds to which of the Results session. Re-organization of the subsections in Methods is recommended. For example, it seems the following lines are all relevant to the scRNAseq data analysis in HUNTER-seq, but the order of procedure and which data preprocessing procedure is for which scRNAseq results are not clear. That are Line 594, 636, 663 and 673. If otherwise, these different sub-sections are for different scRNAseq datasets, it also needs to be clarified.

Overall, a flowchart of the main procedures in Methods may also be helpful.

4, In Line 171-172, it’s not very clear for me why “The AAV8/N-GR administration transduced the N-GR gene primarily into hepatocytes”? Shall we expect it’s only one cell type like hepatocytes will be investigated each time? Is the observation that the transduced N-GR is primarily into on cell type specifically in this example or general? It’s possible that several cell types can interact with the cancer cells.

5, This work is related to inferring CCI in spatial transcriptome (ST). As in discussion, the current may still have limited spatial resolution. ST technology also moves fast. The single cell level of ST (scST) resolution may be expected in the near future. Comparing to scST in a tumor+adjacent tissue, the proposed method requires more wet-lab and model procedures. Can the authors discuss about the future application of HUNTER-seq generally, and the possible ad-vantages in this specific point of view? This will

help readers have a bigger picture of the field of CCI.

Point-by-point response to reviewer comments

We would like to thank all the reviewers for their insightful comments. According to the comments, we have revised the manuscript: text, figure and supplementary figure files. In addition to these files, we have newly provided a supplementary methods file to address the reviewer comments. Our revised text file is a marked-up version, with changes in the text highlighted in blue. The revised manuscript text is also exemplified in our response to each comment.

REVIEWER COMMENTS

Reviewer #1 (Remarks to the Author):

Minegishi et al., have developed a method to fluorescently label in proximal cells in tissue by modifying a technology previous developed in a different lab (Kinoshita at al., 2019 and 2020). Indeed, the corresponding authors of the previous paper are also contributing authors here.

The approach is based on split GFP approach largely used to study protein-protein interaction (review in Romei and Boxer, 2019 10.1146/annurev-biophys-051013-022846).

In Kinoshita at al., 2019 split GFP technology was used by fusing the N-terminal or the C terminal to glycoposphatidylinositol anchor domain and express in two different cells (GRAPHIC). Here GFP signal would reconstitute only if the two expressing cells are in contact with each other. In the present work Minegishi et al., modified this approach to make one of the GFP fragment soluble and not anchored to the plasma membrane allowing for the producing cells to trigger a membrane GFP signal in neighbouring cells expressing the membrane anchored fragment. The author presents a prove of concept for an in vivo application of this approach in the context of hepatocyte-cancer cells interaction during liver metastasis.

In general, the authors present a very neat system, that could lead to the implementation of new genetic mouse models to fluorescently detect cells proximal to specific cells in a tissue. I feel that the manuscript's main value is presentation of this model system with a proof-of-concept validation of its in vivo application. In my opinion, the work is in principle suitable for publication in Nature Communication. However, the current version suffers for some data overinterpretation, more details need to be provided for the in vivo validation and some part of the text need to be revised.

Major comments on state-of-the-art contextualization and objectivity

1. There are certainly many discoveries to be made in the cancer field and so much we still do not know, but the notion of the “metastatic niche” formation it is not one of them. When typing “metastatic niche” in pubmed, there are currently 1,854 studies only in the last 5 years. While it is a fact that how such complex phenomenon is governed or tailored to different cancer cells it is subject of intense investigation, I found rather unacceptable how the authors start the manuscript (abstract and introduction) in a way that put in doubt the very existence of such well known phenomenon. The authors use sentences like “hypothetically establishing a metastatic niche”, “metastatic niche hypothesis”, “...cancer cells hypothetically form metastatic niche”.

This reviewer agrees that a great limitation in studying the metastatic niche are tools that allow to detect tissue cells proximal to metastatic cancer cells, but we certainly do not need those tools to know that the metastatic niche exist.

The manuscript should be written in a way to objectively describing the advantages of the sGRAPHIC in the context of the knowledge of the field.

Answer: Thank you for your comment to encourage the significance of the metastatic niche. We have revised the sentences (lines 36-37 and 56) to push the assumption of the metastatic niche though some researchers still disrespect metastatic niche hypothesis in cancer research filed.

(lines 36-37) Cancer cells inevitably interact with neighboring host tissue-resident cells during the process of metastatic colonization, **establishing a metastatic niche to fuel their survival, growth, and invasion.** However, the underlying mechanisms in the metastatic niche is yet to be fully elucidated owing to the lack of methodologies for comprehensively studying the mechanisms of cell–cell interactions in the niche.

(line 56) During metastasis, **cancer cells form metastatic niches** to gain benefits for their growth through interactions with neighboring tissue-resident cells during metastatic colonization processes

2. On the similar line, when presenting the split GFP strategy both advantages and limitation need to be clearly described in order to avoid too biased messages. For examples, when comparing the efficiency of sGRAPHIC with the Cherry-niche system the authors clearly show that the reconstituted GFP signal is much brighter compared to the internalized mCherry signal in recipient cells, and that in vitro, sGRAPHIC can labelled a large proportion of cells when

expressed in this particular cell line. In the discussion session, the authors do contextualize the result to the cell type used, but they also need to add that conclusion on the comparison of the two system cannot be fully made, as they were not compared in the most relevant *in vivo* approach. Moreover, the authors also need to discuss that the main difference in the two system: namely that in the sGRAPHIC both recipient and donor cells needs to be genetically modified. Therefore, to achieve an unbiased detection of the various cell surrounding cancer cells *in vivo*, the sGRAPHIC will require the generation of a N-GR transgenic mouse model under a ubiquitous promoter, like Actin promoter. I believe that generating the mice would not be required for publication, but it should be at least mentioned in the discussion.

Answer: We appreciate your comment that improves the integrity of our manuscript. To clarify pros and cons of sGRAPHIC approach, we have discussed them on lines 314-329.

(lines 314-329) In this light, the Cherry-niche system was a successful system to identify metastatic niche cells through fluorescent labeling with secretory mCherry proteins. Although the *in vivo* comparison between Cherry-niche and sGRAPHIC is still lacking in this study, we reproduced Cherry-niche labeling with HEK293T cells, and showed that the fluorescent labeling efficiency of sGRAPHIC surpassed that of Cherry-niche (Figs. 1c - e, and S4). In addition, Cherry-niche has not been detailed for its fluorescent labeling efficiency and applicability in various cell types. In contrast, sGRAPHIC archived fluorescence labeling of multiple types of cell–cell interactions with high efficiency (Figs. 2a, b, S5, and S6). These results demonstrate that sGRAPHIC enables for optical labeling of transient cell–cell interactions in broad biological contexts. While, sGRAPHIC requires gene transduction of the reporters in both cancer cells and tissue-resident cells, whereas Cherry-niche requires the reporter only in cancer cells. As we employed, AAV-mediated gene transduction is a speedy strategy, but limited types of cells are targetable (Fig. S10). To overcome this shortcoming, transgenic mice ubiquitously expressing N-GR reporter are desired for targeting diverse tissue-resident cells interacting with cancer cells.

Major comments on presented data

The *in vitro* experiments describing the sGRAPHIC system are overall well-presented and described. The main concerns are around the *in vivo* validation.

1. Given the fact that by using the adenovirus delivery of the N-GR construct, the data show that cells targeted are hepatocyte (Fig S10), the fact that in this prove of concept that sGRAPHIC can

detect tissue cells surrounding cancer cells in the tissue is limited to hepatocytes.

Answer: The reviewer correctly pointed out a limitation in the proof of concept of *s*GRAPHIC labeling in this study. We have demonstrated here that *s*GRAPHIC labeling is successful in advancing our understanding of cell–cell interactions, even limited in hepatocytes, neighboring on metastatic colonies. However, previous studies have suggested that other types of tissue-resident cells are involved in liver metastatic niche mechanisms (Proc Natl Acad Sci 2010, 107, 13063, Nat Commun 2021, 12, 863). Although these studies investigated the mechanism of metastatic niche in colorectal cancer models, other cell types in addition to hepatocytes are likely involved in our breast cancer metastatic niche model. A major limitation of AAV-based reporter gene delivery is that AAV generally targets non-dividing cells in living tissues. In the liver, the majority of AAV-targeted cells are reported to be hepatocytes (Gene Therapy 2003,10, 2105, PLoS ONE 2023, 18, e0283996), as we have shown in our manuscript (Fig. S10). To maximize the application of *s*GRAPHIC, N-GR transgenic mice are required. We have discussed this point on lines 324-329 of the revised manuscript.

(lines 324-329) While, *s*GRAPHIC requires gene transduction of the reporters in both cancer cells and tissue-resident cells, whereas Cherry-niche requires the reporter only in cancer cells. As we employed, AAV-mediated gene transduction is a speedy strategy, but limited types of cells are targetable (Fig. S10). To overcome this shortcoming, transgenic mice ubiquitously expressing N-GR reporter are desired for targeting diverse tissue-resident cells interacting with cancer cells.

2. Given the strong GFP signal shown in the co-culture as well as the bright GFP signal in the liver metastasis in Fig 2c, can the author explain why, in the FACS plot of figure S12 the population of GFP labelled hepatocytes is hardly visible? Can the authors provide a quantification of the ratio of Azurite+ cancer cells vs GFP+ hepatocyte in liver metastasis?

Additionally, when comparing the GFP signal of GFP- hepatocytes to the one of healthy hepatocyte, it looks like there is a shift up in the overall population (Fig S12), can the author provide an explanation for this? Can the author estimate the distance of GFP labelling from the producing cancer cells?

Answer: Thank you for pointing out a very important issue. We believe that the data and discussion presented below, which have also been added to the revised manuscript, adequately address all questions raised by the reviewers.

Why GFP-positive hepatocytes are hardly visible in FACS plots? & Shift up of FACS plots in liver metastasis sample

Answer: Hepatocytes are very fragile cells. Therefore, we must perform rapid perfusion under gentle enzymatic dissociation (collagenase supplemented with trypsin inhibitor) to efficiently isolate living hepatocytes. The number of hepatocytes, which occupy 80% of the liver volume, is estimated to be $>10^8$ cells (/20 g body weight) in a mouse (In Vitro 1981, 10, 913), but in reality, only about $<10^7$ cells (/liver) of hepatocytes can be harvested in the current standard protocol with collagen-perfusion and centrifugation-based isolation. The proportion of hepatocytes adjacent to a small number of metastatic foci would be considered a very small fraction of the total tissue. In addition, accumulation studies have shown that inflammation often triggers cell death programs in hepatocytes (Hepatology 2004, 39, 273, J Clin Invest 2017, 127, 55). Our scRNA-seq analysis also suggests that the GFP-positive hepatocytes are likely to be under inflammatory stressful conditions (Figs. 3b, S14 and S15). It is quite possible that even fragile hepatocytes are even more susceptible to cell death during stressful dissociation process, reducing the isolation yield in flow sorting. Therefore, we tried to concentrate GFP-positive hepatocytes by local mechanical destruction after collagenase-perfusion. Mechanical destruction of whole liver tissue has been standardly employed after collagenase perfusion in the two-step collagenase perfusion procedure. We isolated GFP-positive cells from the liver harboring overt metastatic colonies that have a few mm in a diameter, and easily visible through the naked eye (Fig. S12a). After collagenase perfusion, we selectively destructed visible metastatic colonies and surrounding hepatic tissues to increase GFP-positive hepatocytes in total population. Consequently, we found that the increased GFP signals in the liver metastasis sample compared with the control sample and then gated GFP-high population as the GFP-positive proximal hepatocytes (Fig. S12b).

The local destruction process resulted in that total population mainly included the hepatocytes located in the area relatively close to the metastatic colonies in the liver tissue. We assumed that most of these hepatocytes were susceptible for sGRAPHIC labeling, resulting in the shift-up of the whole population in FACS analysis (Fig.S12b). Similarly, the shift-up of whole population was also found in FACS analysis of co-culture with E0771/sC-GR and NIH3T3/N-GR at a cell number ratio of 1:50 (Fig. S7) even though microscope observation showed that sGRAPHIC labeling extended across only one-two cell layers from cancer cells (Fig.2b). This suggests that the range of sGRAPHIC labeling has a long tail that is detectable in FACS but hard to be visualized in microscopic observation. The range of *in vivo* sGRAPHIC labeling was found to be significantly wider than that of *in vitro* labeling: $99.7 \pm 13.4 \mu\text{m}$ (Fig. S11d) versus $56.7 \pm 6.8 \mu\text{m}$ (calculated from images (n=4) of co-culturing E0771/sC-GR and NIH3T3/N-GR in 1:50 ratio). This fact further evidence that sGRAPHIC labeling might cause the shift-up of the whole population of our hepatocyte isolation from the metastasized liver tissues in FACS analysis.

In the revised manuscript, we have detailed the method on hepatocyte isolation (lines 673-675) and provided discussion on *s*GRAPHIC labeling range in various settings (lines 285-300).

(lines 673-675) Three weeks after cancer cells transplantation, primary hepatocytes and cancer cells were harvested from dissociated murine livers through the two-step collagenase perfusion procedure. In this step, we selectively destructed an area with a visible metastatic colony to increase yields of GFP-positive hepatocytes. The harvested cells were fluorescently sorted into 96-well plates for scRNA-seq with BD FACSAria™ (Becton, Dickinson and Company).

(lines 285-300) While, *s*GRAPHIC labeling in the liver metastasis tissue occasionally extended across several cell layers from cancer cells, which appeared to be longer than in the *in vitro* settings (Figs. 2b, d and S11d). This may be explained by the efficient diffusion of secretory GFP fragments in narrow stromal space of living tissues. At the same time, there are several potential factors in living tissues that influence the diffusion of GFP fragments besides the Brownian motion that governs the fragment diffusion in the *in vitro* culture system. For example, high-density proliferation of cancer cells often increases intra-tissue pressure, resulting in a gradient of interstitial flow to surrounding healthy tissue³⁰. In addition, interactions between secretory proteins and the extracellular matrix are reportedly crucial to define the diffusion of morphogens in developmental tissues³¹. Cancer cells heterogeneously produce extracellular matrices in the lesions³². These facts probably explain varied range of *s*GRAPHIC labeling in the *in vivo* setting (Figs. 2c and d). Therefore, we could control *s*GRAPHIC labeling range in living tissues through understanding the interactions between secretory GFP fragments and extracellular tissue components.

Quantification of the ratio of Azurite+ cancer cells vs GFP+ hepatocyte in liver metastasis

Answer: We faced difficulties in obtaining a biologically meaningful number ratio of cancer cells to GFP-positive hepatocytes. Effective dispersion of metastatic colonies requires prolonged and more intense enzymatic treatment, but harvesting living hepatocytes under the same conditions was difficult due to their fragility. Furthermore, when blood flow in the liver is unstable due to accidental thrombus formation or other factors, perfusion is partially unsuccessful, resulting in the unsuccessful isolation of hepatocytes from specific areas. Local inflammation and unstable blood flow may well occur when liver metastases form. For these reasons, it is difficult to stably detect the biologically meaningful number ratio of cancer cells to hepatocytes in liver tissue by FACS.

Estimation of the distance of GFP labeling from cancer cells in vivo

Answer: In the revised manuscript, we added new imaging data for *s*GRAPHIC labeling *in vivo* (Figs. 2c, d and S11c, d). To obtain these data, we changed the experimental site and microscope from the those in the original manuscript (from RIKEN with FV3000 to Jichi Med Univ with SP-8). Because it is difficult to compare images obtained with different microscopes, we excluded the imaging data of Fig. 2c in the original manuscript.

During this manuscript revision work, we realized that *in vivo s*GRAPHIC labeling occasionally extended across several cell layers and the maximum range of the labeling was calculated to be $99.7 \pm 13.4 \mu\text{m}$ (Fig. S11d). To calculate the range of labeling, we measured the distance between the nuclei of the furthest GFP-positive hepatocytes and the nuclei of the most marginal cancer cells in confocal fluorescence images with relatively well-defined borders in the metastatic colony–hepatic tissue. Interestingly, the distance of *s*GRAPHIC labeling in the invasive borders (Fig. 2c) is probably shorter than that in metastatic colonies with well-defined borders (Fig. 2d). We have not identified the molecular mechanisms behind this observation, but we discussed a potential mechanism from a point of view of the interactions between secretory GFP fragments and the extracellular matrix that is often heterogeneously accumulated in tumor tissues and accelerates cancer cell invasion (lines 294-300).

(lines 294-300) In addition, interactions between secretory proteins and the extracellular matrix are reportedly crucial to define the diffusion of morphogens in developmental tissues³¹. Cancer cells heterogeneously produce extracellular matrices in the lesions³². These facts probably explain varied range of *s*GRAPHIC labeling in the *in vivo* setting (Figs. 2c and d). Therefore, we could control *s*GRAPHIC labeling range in living tissues through understanding the interactions between secretory GFP fragments and extracellular tissue components.

3. In the *in vivo* setting the healthy liver refers to mice that did not received cancer cells, but where the mice injected with the adenovirus to label the hepatocytes? Of course, this would have been the correct control to use and based on Fig S12, as the hepatocytes express Cherry, I assume this was the case. However, Fig 4a show that the main difference in hepatocyte signature is the presence of cancer cells in the liver, compare to that, the difference between hepatocyte near or far from the cancer cells is much less evident. Is this possible? In absence or primary tumour, when injecting cancer cells directly into the tissue, why would hepatocyte far from seeding cancer cells are so deeply influenced? At an early stage of metastatic colonization, this is very unlikely. To induce such dramatic whole tissue effect the metastatic load would need to be very high, how big were the metastasis when the livers were analysed? Can the authors provide an explanation for this data?

Answer: As the reviewer pointed out, the healthy mice were injected with adenovirus, and not implanted with cancer cells. To clarify this point for readers, we have added a schematic diagram in Fig. S12.

As described in our response to comment 2, we harvested hepatocytes that locate relatively close to metastatic colonies. A previous study has reported that cytokines released into the bloodstream from the primary tumor can alter the phenotype of hepatocytes (Nature 2019, 101, 2011). Although the amount of cytokines secreted from metastatic foci is smaller than that from the primary tumor, it is likely to have a greater impact on hepatocytes due to the short distance between cells in the same tissue.

4. The authors provide a very general analysis of the single cell sequencing data, which is fine as a proof of concept, but they also need to acknowledge that mechanistic conclusions cannot be made without a functional validation. The message that the expression of Galectin-3 found in hepatocyte in the niche can supporting cancer cells growth is supported by a functional validation at least *in vitro*. But there is no data showing that Galectin-3 in hepatocytes is important for liver metastasis *in vivo*, therefore, conclusion need to be tone down. To determine Galectin-3 requirement *in vivo*, the authors could use Adenovirus to deliver shRNA to KD Galectin-3 to the hepatocyte to show than *in vivo* this leads to reduce metastatic efficiency.

Answer: As the reviewer suggested, KD of galectin-3 in hepatocytes would be crucial to solid our biological findings. However, *in vivo* KD using an adenovirus-associated vector requires very tough validation of the KD system both *in vitro* and *in vivo*, and we are still on going to optimize the experimental condition. The demonstration of the mechanism is an additional part of this paper, although we believe it is important. Therefore, in this revision, we have given up on presenting data of *in vivo* KD, and have excluded the summary diagram (Fig. S16 in the original manuscript) to avoid overstatement. In addition, we have softened the concluding sentences in Abstract (lines 47-48) and Results sections (lines 245-246), and revised the Graphical Abstract.

(lines 46-48) Among the marker genes of MAHs, we identified *Lgals3*, encoding Galectin-3, as a potential pro-metastatic factor that accelerates metastatic growth and invasion.

(lines 245-246) Overall results implicate that Galectin-3 is a pro-metastatic factor that mediates interactions between cancer cells and hepatocytes in liver metastatic lesions.

5. No conclusions should be made on functional value of Myc activation in hepatocyte. Unless a

similar KD Myc in hepatocyte approach using adenoviruses suggested above is made to show its requirement *in vivo*. Without this functional evidence, the Myc activation should not be presented in the model in Fig S16 and summary image.

Answer: We have excluded the summary diagram (Fig. S16 in original manuscript) in the revised manuscript. The network analysis including *Myc* (Fig. S14) remains for discussion on the gene regulatory mechanism in MAHs.

6. Similarly, no conclusions should be made on the induction of Myc via IL2, the fact that it is found expressed in cancer in the liver certainly does not provide a causative link for Myc activation in the hepatocyte. This data can be maintained in the discussion as a speculative possibility but must be removed from the model in Fig S16 and summary image.

Answer: We have excluded the summary diagram (Fig. S16 in original manuscript) in the revised manuscript. In the revised manuscript, supplemental data on IL2 was used for discussion purposes only.

I personally do not think that the authors need to provide necessarily a novel mechanistic inside a about liver metastasis, but the data of the *in vivo* labelling need to be clearer, with more imaging of labelling, distance from the cancer cells, metastatic size and cancer cell to labelled hepatocyte ration.

Answer: Thank you for your appreciation of the concept of *sGRAPHIC*. To validate *in vivo* *sGRAPHIC* labeling, we have revised the manuscript by adding new imaging data (Figs. 2c, d and S11c, d). We have also provided detailed information for liver metastasis sample in FACS analysis (Fig. S12) and the protocol for harvesting GFP⁺ cells. For these revised data, we have reorganized the sections of Results and Discussion to improve the integrity of the manuscript. We believe that these revisions would sufficiently deliver the concept of *sGRAPHIC* and its application to metastasis research field.

Reviewer #2 (Remarks to the Author):

(Please see also attached pdf with additional comments/notes)

Key results

The authors of the manuscript built upon an existing and previously published optical labelling system GRAPHIC to make an improved version of it called sGRAPHIC to specifically study cell-cell interactions in metastatic niches. They combined sGRAPHIC with single-cell RNA sequencing thus creating the HUNTER-seq platform. This platform was used to study the gene expression patterns in metastatic niche-associated hepatocytes (MAHs) in a murine model of liver metastasis. Galectin-3 expression was found to be upregulated in those MAHs. Galectin-3 is known as a pro-metastatic factor.

Validity

I would like to see a clearer description of differences of sGRAPHIC from a previously published GRAPHIC system. In what way is the system new and improved?

Answer: We have revised the beginning part of Results section (lines 95-108) and added new fluorescence images (Figs. S1b and S2e) to clearly show the different applicability of GRAPHIC and sGRAPHIC. We have also added a discussion of GRAPHIC vs sGRAPHIC in lines 258-273.

(lines 95-108) We tested the applicability of the GRAPHIC system in cancer cell lines by genetic transduction of C-GRAPHIC (C-GR) or N-GRAPHIC (N-GR) reporter. These reporters express cell membrane-anchored C- or N-terminal GFP fragments. To distinguish cells expressing N- or C-terminal GFP fragments by a fluorescent marker of the nucleus, the reporters also encode fusion proteins of histone H2B protein, and red fluorescent protein mCherry or blue fluorescent protein Azurite (Fig. S1a). The GRAPHIC system efficiently labeled cell-cell interactions in the epithelial cell line LLC-PK1 cells as we previously demonstrated¹¹, but the system was not functional in the cancer cell line HeLa cells (Fig. S1b). We speculated that the inefficient fluorescence labeling of GRAPHIC in cancer cells was due to the unstable cell-cell adhesion of cancer cells. To achieve efficient optical labeling of cell-cell interactions involving cancer cells, sGRAPHIC was conceived using a combination of GPI-anchored N-terminal and secretory C-terminal GFP fragments (Fig. 1a).

(lines 258-273) However, GRAPHIC is unable to fluorescently label cell–cell interactions involving cancer cells, probably because the highly migratory nature of cancer cells does not allow reconstituted GFPs to be maintained between cells (Supplementary Videos 2 and 3). Even when cancer cells statically form cell–cell junctions, the membrane-anchored GFP fragments are not efficiently reconstituted in an unstable intercellular space due to heterogeneous cell–cell junction proteins²². The distance between cell–cell membranes at intercellular junctions is highly variable because the extracellular domains of cellular junction proteins are range from several nm at the tight junction to over 100 nm at the desmosomal junction^{23, 24, 25}. The polypeptide linkers used to display GFP fragments on the cell membrane are perhaps capable of bridging the distance of 30 nm, but apparently not 100 nm. Extending the extracellular linker of GRAPHIC is a potential strategy to overcome the variable distance between cells, although this strategy may not be effective in overcoming the problem of the high mobility of cancer cells. sGRAPHIC successfully overcomes the shortcomings of GRAPHIC, and expands the applications of split-GFP-based labeling systems for cell–cell interactions.

Second, power analysis used as part of statistical analysis to estimate the necessary sample size(s) to look for significant differences?

Answer: We did not perform a power analysis to determine the sample size for each experiment. We conducted preliminary experiments empirically to determine the sample size needed to detect significant differences in the means.

Significance

The manuscript describes in great detail a new and very useful application of the previously published optical labeling system combined with additional types of analysis, like flow cytometry and scRNA-seq, to study cell-cell interactions more effectively in metastatic niches, combined with the study of gene expression changes. Use of cells of different cancer types allows for wider application of the proposed sGRAPHIC system.

The previously published system is described in Kinoshita, N. et al. Genetically Encoded Fluorescent Indicator GRAPHIC Delineates Intercellular Connections. *iScience* 15, 28-38 (2019). <https://doi.org:10.1016/j.isci.2019.04.013>

Answer: Thank you for your appreciation of the concept of sGRAPHIC. As we answered for the comment of “Validity”, the revised manuscript details sGRAPHIC versus GRAPHIC to clarify the differences of these systems.

Data and methodology

Overall, in my assessment, the used approach is valid, the quality of all the data including the supplemental materials is sufficient, and the quality of presentation can be improved in terms of clarity.

Answer: Thank you for the comment. To answer the comment³ from reviewer³, we have revised methodology sections to clarify the link between a method and an experiment in HUNTER-seq (lines 710-743).

Analytical approach

Since I am not an expert in optical labeling systems, It is hard for me to evaluate the validity of analytical approach in this regard. I would like to know if power analysis was used as part of statistical analysis to estimate the necessary sample size(s) to look for significant differences?

Answer: We did not perform a power analysis to determine the sample size for each experiment. We conducted preliminary experiments empirically to determine the sample size needed to detect significant differences in the means.

Suggested improvements

I have included a number of suggested improvements in my comments added to the pdf of the manuscript (uploaded). I think the manuscript present a solid piece of existing systems and technology application with valid and useful results, but improvements are needed in manuscript clarity.

I highly recommend adding a list of used abbreviations.

Answer: Thank you for all your comments. We have answered each comment point-by-point to each comment in the attached pdf file. We have also added a table of abbreviations to the Supplementary Methods file that is new in the revised manuscript.

Clarity and context

In my view, the clarity of the text can be improved, with adding the explanations why the presented types of analysis were used and what is innovative or novel in the used approaches. I agree that the results have been provided with sufficient context and consideration of previous work.

Answer: Thank you for your comments. We believe that the integrity of the manuscript has been improved through this revision.

References

The manuscript references previous literature appropriately.

Answer: Thank you for your comment. We have added new references for revised texts, and believe that they are appropriate.

My expertise

I don't have expertise in constructing, using or evaluating optical labeling systems.

Answer: We believe that the revisions made in line with your comments will improve the readability of our manuscript for a wide range of readers. Thank you for all your valuable comments.

Reviewer #3 (Remarks to the Author):

This paper is to develop the sGRAPHIC system as a new genetic optical labeling system, based on previous GRAPHIC system by adding secretary-C terminal GR (sC-GR, I can't find what's GR for in this paper) that can bind the N- terminal GR (N-GR) to activate green florescence and use it to understand the cell-cell interaction (CCI) between metastasized cancer cells and different tissue-resident cells. Several aspects of the development of this technique have been evaluated and optimized, including the pair of sC-GR and N-GR, the half-life of reconstituted GFP, labelling performance both in vitro and in vivo. Particularly, HUNTER-seq was further developed as application of sGRAPHIC combined with the single cell RNAseq (scRNAseq) technology to sequence the obtained fluorescent neighbor tissue-resident cells near to metastasized liver tumor cells (from breast cancer cell line originally). Candidate genes and pathways have been identified with promising interpretations based on

previous literature. This is a very intriguing novel procedure to study an important question of CCI between metastasized cancer cells and different tissue-resident cells that may provide microenvironment of metastasis. The paper also described and discussed the limitation of previous methods. Overall, the work sup-ports the conclusion.

The paper is carefully well-written. However, there are spaces to improve this work in terms of the structure and the content for clarification of the developed method and reproducibility.

Major comments (mostly in data analysis):

1, In Line 193-194, three groups of hepatocytes (1) proximal, (2) distal and (3) from healthy controls. In Line 195 "Differential gene (DE) expression analysis", it's not clear whether DE genes were obtained by comparing the pooled proximal and distal hepatocytes (from metastasized samples) to the hepatocytes in the control group. If so, the later methods or this section need to clarify it. In Methods, "analysis of variance" in Line 670 and "Rank Test" in Line 681 were mentioned, but neither seems exactly reflects the DE analysis here.

Same question about sample group comparison remains in Line 199-201 for gene ontology analysis. Is it two group comparison or paired of three group comparisons? What (logFC? proximal vs each of the other two or pooled of the other two?) are based to rank genes?

Answer: As the reviewer pointed out, differential expression analysis for Figure S13 was done

by comparing the pooled proximal and distal hepatocytes to control hepatocytes. To clarify this point, we have revised a sentence (lines 210-212) and integrated the relevant methods into “Data analysis for HUNTER-seq” (lines 710-743). Similarly, we have revised the relevant sentence to clarify the grouping in gene ontology analysis (lines 215-217). In addition, we have revised the figure legends (Figs. 3b, 4b and S13) to provide the detail of DE expression analysis.

We agree that lines 670 and 681 in the original manuscript incorrectly described the methodology. To improve these points, we have revised the text on lines 710-743.

We also have replaced with the data of GO analysis with those obtained by the latest version of program packages (Fig. 3b and S15a). We detected some changes in p-value of ontology terms, but did not add any additional discussion because the biological conclusions were unchanged from the original manuscript. The version of the program packages (Seurat, clusterProfiler, org.Mm.eg.db) has been provided on lines 717 and 729 in the revised manuscript.

(lines 210-212) Differential gene expression analysis detected the upregulation of *Saal* and *Lcn2* in hepatocytes (pooled proximal and distal) from the metastasized liver compared to those (control) from the healthy liver (Fig. S13).

(lines 710-743) **Data analysis for HUNTER-seq.** The raw reads were preprocessed with UMI-tools (1.0.0)⁵⁸ and demultiplexed by *fqtools* (2.1)⁵⁹ and *subseq* of *seqtk* (1.3-r106) (<https://github.com/lh3/seqtk>). Input reads were down-sampled to be 23,918 reads per cell with *seqtk*. Mapping of sequence reads to a reference genome (GRCm38) was done with STAR (2.7.9a)⁶⁰. The aligned reads were annotated by *featureCounts* (2.0.1)⁶¹ with *Mus_musculus.GRCm38.102.gtf*. The cell barcodes and unique molecular identifier (UMI) were quantified with UMI-tools (1.0.0)⁵⁸. Single-cell sequence data were analyzed using the Seurat R package (4.3.0)^{62,63}. We filtered out cells with more than 7% mitochondrial gene expression or fewer than 3000 unique transcripts from the analysis. The counts were normalized with a log-normal transformation and scaled with a scale factor of 100,000. The *VlnPlot* function in Seurat was used for visualizing expression levels of genes of interest. Differentially expressed genes (DEGs) among two groups were identified using the *FindMarkers* function and the Wilcoxon Rank Sum test in Seurat by thresholding at defined adjusted P-value and Log2 fold change (FC) (Figs. 3b, S13 and S15a). The marker genes of clustered populations were identified by comparing the cells in a population with all other cells with the *FindAllMarkers* function with the Wilcoxon Rank Sum test in Seurat (Figs. 4a, 4b and S14). Functions enriched with the up- or down-regulated genes (DEGs with positive and negative Log2FC) were respectively analyzed by the GO functional annotation in clusterProfiler (4.6.0)⁶⁴ on org.Mm.eg.db (3.16.0)⁶⁵ (Fig. 3b). S and G2-M phase scores were calculated using the *CellCycleScoring* in Seurat with mouse cell cycle phase

genes (https://github.com/hbc/tinyatlas/blob/master/cell_cycle/Mus_musculus.csv). S and G2-M phase scores were statistically compared among proximal, distal and control hepatocytes with analysis of variance and Tukey honestly significant difference test and visualized by the *VlnPlot* function in Seurat (Fig. 3c). The PCA on the single-cell expression matrix was performed with the *RunPCA* function and the marker genes for hepatocyte groups (proximal, distal, and control). Clustering of hepatocytes was performed the *FindNeighbors* and the *FindClusters* function. t-Distributed stochastic neighbor embedding (t-SNE)⁶⁶ was used for visualization (Fig. 4a). The proximal and distal hepatocytes were ordered in Fig. 4b by the *DiffusionMap* function in *destiny* (3.12.0)⁶⁷ with the top 20 upregulated genes in clusters 2 and 3, and the levels of gene expressions were visualized by using CRAN *pheatmap* (1.0.12) (Fig. 4b). Gene regulatory networks of cluster 3 were analyzed by SCENIC (1.2.4)^{68,69} (Fig. S14).

(lines 215-217) Gene ontology analysis of marker genes expressed in proximal hepatocytes over distal and control hepatocytes highlighted that proximal hepatocytes were characterized by stress responses and loss of liver metabolic functions (Fig. 3b).

(Figure 3b legend) Gene Ontology of up-regulated (top) and down-regulated (bottom) genes (adjusted P-value < 0.1, Log 2 FC absolute value > 0.3) in the proximal hepatocytes as compared to the distal and control hepatocytes.

(Figure 4b legend) Heatmap of cluster 2 and 3 marker genes of the top 20 ranked by Log 2 FC (adjusted P-value < 0.05) expressions in the GFP+ proximal hepatocytes.

(Figure S13 legend) Scatter plot of differential gene expression between hepatocytes from the metastasized (proximal and distal) and those from the healthy livers (control). The vertical dot lines correspond to a log 2-fold change of -0.27 and 0.27, while the horizontal dot line indicates an adjusted P-value of 0.5.

2, The caption in Figure S14 needs more details. What comparisons are the “up-regulated gene” from? How many of them? Are all of them connected as in S14? How correlation was calculated in SCENIC and what’s the cutoff of correlation for edges in the network? Some related details are in Methods, but not quite sufficient to allow clear understanding.

Line 705-706, “Highlighted gene regulatory
706 networks in cluster 3 with CoExWeight more than 0.007”. It’s not clear what does this mean. Overall, the procedure of network analysis needs to be further clarified.

Answer: We have revised the legend of Fig. S14 to provide the details in the gene regulatory network analysis.

(Figure S14 legend) Gene regulatory network (GRN) around *Myc* in hepatocytes of cluster 3 defined in Fig. 4a. GRNs were inferred by single-cell regulatory network inference and clustering (SCENIC, 1.2.4^{68, 69}) workflow. The resulted GRN of cluster 3 was visualized with marker genes (418 genes) satisfying adjusted p-value < 0.01, average log 2-fold change > 1, CoExWeight > 0.007 with *Myc*, *Klf2*, *Tfdp1*, *Sap30*, *E2f4* using CRAN igraph (1.3.2). The transcription factors and marker genes of cluster 3 listed in Fig. 4b were highlighted with magenta and neon green, respectively.

3, The Methods section is a bit difficult to follow. It's not always easy to tell which method procedure corresponds to which of the Results session. Re-organization of the subsections in Methods is recommended. For example, it seems the following lines are all relevant to the scRNAseq data analysis in HUNTER-seq, but the order of procedure and which data preprocessing procedure is for which scRNASeq results are not clear. That are Line 594, 636, 663 and 673. If otherwise, these different sub-sections are for different scRNAseq datasets, it also needs to be clarified.

Overall, a flowchart of the main procedures in Methods may also be helpful.

Answer: To improve the flow of the Methods section, we have reorganized the several paragraphs in the Method. First, we newly prepared a Supplementally Methods file to separately describe the experiments mainly presented in Supplementally Figures. Especially, we have integrated the method of scRNA-seq for liver cells (Fig. S10) into "Analysis of gene transduction into liver cells" in the Supplementally Methods file. Second, we have integrated small paragraphs into "Data analysis for HUNTER-seq" (lines 710-743), and indicated the corresponding figs for each analysis. In addition, we have provided detailed conditions for HUNTER-seq analysis in the figure legends for clarity (Figs. 3b, 4b, S13 and S14).

4, In Line 171-172, it's not very clear for me why "The AAV8/N-GR administration transduced the N-GR gene primarily into hepatocytes"? Shall we expect it's only one cell type like hepatocytes will be investigated each time? Is the observation that the transduced N-GR is primarily into on cell type specifically in this example or general? It's possible that several cell

types can interact with the cancer cells.

Answer: As the reviewer pointed out, the applicability of *s*GRAPHIC will be maximized if N-GR can be expressed in all types of tissue resident cells. However, this would require transgenic animals expressing N-GR systemically, which would take time to generate. On the other hand, AAV is a ready-to-use method for transducing the gene into tissue resident cells, but the reporter gene is maintained only in non-dividing cells, which limits the type of target cells for *s*GRAPHIC labeling. In this study, we used the AAV-based approach as a proof of concept for the *in vivo* application of *s*GRAPHIC. To discuss this point, we have provided additional discussed on lines 324-329.

(lines 324-329) While, *s*GRAPHIC requires gene transduction of the reporters in both cancer cells and tissue-resident cells, whereas Cherry-niche requires the reporter only in cancer cells. As we employed, AAV-mediated gene transduction is a speedy strategy, but limited types of cells are targetable (Fig. S10). To overcome this shortcoming, transgenic mice ubiquitously expressing N-GR reporter are desired for targeting diverse tissue-resident cells interacting with cancer cells.

5, This work is related to inferring CCI in spatial transcriptome (ST). As in discussion, the current may still have limited spatial resolution. ST technology also moves fast. The single cell level of ST (scST) resolution may be expected in the near future. Comparing to scST in a tumor+adjacent tissue, the proposed method requires more wet-lab and model procedures. Can the authors discuss about the future application of HUNTER-seq generally, and the possible ad-vantages in this specific point of view? This will help readers have a bigger picture of the field of CCI.

Answer: We agree with the reviewer's perspective that the spatial resolution of tissue-section based ST will reach single-cell resolution in the near future. However, as discussed in the original manuscript, one problem with the current tissue-based ST is low sequence depth. It may be somewhat uncertain whether it will be possible in the future to improve sequence depth while at the same time improving spatial resolution. Another disadvantage of tissue-section based ST is its throughput. To identify the section containing the tiny cell population of interest, it is necessary to go through sequential sections in the organs. In contrast, HUNTER-seq can extract cell-cell interactions from organs in a single step. In addition, the isolation of living cells in HUNTER-seq allows us to perform conventional live cell assays to further mechanistic insights. We have additionally discussed this point in lines 359-367 in the revised manuscript.

(lines 359-367) In addition, to identify the section containing the tiny cell population of interest, it is necessary to go through sequential sections in the organs. In this regard, *s*GRAPHIC allows for deep sequencing of defined single cells harvested from the whole tissues using the HUNTER-seq platform, thus addressing the shortcomings of tissue section-based transcriptomics. Furthermore, by isolating living cells of interest, the analytical capabilities of conventional live cell assays can be maximized. More importantly, connecting transcriptome with emerging single-cell multi-omics technologies, including genomics, epigenomics, proteomics, and metabolomics, would be a key to mapping a broad range of cellular statuses ^{52, 53, 54, 55}.

REVIEWER COMMENTS

Reviewer #1 (Remarks to the Author):

I am pleased to see that the authors have made the changes and addressed my previous comments. So in principle for me the manuscript can be published. Just one last small improvement should be made: I understand the authors have decided to tone down the conclusions of Myc and Gal3 as opposed obtain an in vivo validation. However, I feel that at least a validation of the presence of Gal3 protein in vivo in the metastasis area vs distal liver is needed as a quantification. Fig 4c, shows Gal3 expression in the RNA seq data and Fig 4d shows a representative IF staining, however, this should be shown alongside a quantification of this IF staining from different imaging, which properly can confirm the RNA data, before moving to Fig 4e-f where Gal3 is tested on cancer cells ex vivo.

Also the following sentence in the discussion is unclear “In addition, Cherry-niche has not been detailed for its fluorescent labelling efficiency and applicability in various cell type”. The use of that model was published using breast cancer but also leukaemia cells (Passaro et al., Cell Rep. 2021), wouldn't this show it in very different cells, or the author means something else by this? The sentence is not clear.

Reviewer #2 (Remarks to the Author):

After conducting a thorough examination of the revised version of the manuscript, I am pleased to note that the authors have comprehensively addressed the comments provided by all the reviewers, including mine. The authors' responsiveness and attention to the reviewers' feedback are commendable, as they have taken the necessary steps to ensure the manuscript's improved quality and validity.

Notably, the authors not only revised the text according to the reviewers' suggestions but also made significant improvements by incorporating additional figures. These additions have improved the visual representation of the research findings, enhancing the overall clarity of the manuscript.

However, one area that still requires attention is the graphical abstract. It still falls short of delivering a clear and concise message about the research. A graphical abstract serves as a visual summary of the study's key findings and implications, aiming to capture the reader's attention and provide a quick overview of the research content. Therefore, I suggest further refining the graphical abstract to ensure that it effectively communicates the main highlights of the research.

Overall, the manuscript's revisions and additions demonstrate the authors' diligence in addressing the reviewers' comments and improving the overall quality of the study.

Reviewer #3 (Remarks to the Author):

Generally, all of my comments have been well-addressed. I have the following minor points to be further clarified.

LogFC from DE has been used as one of the several criteria in a few figures/results.

Specifically, in Line 939, Figure 4b and Figure S14, is “log2 FC” or “absolute value of log2 FC” used as the criteria?

For Figure 4b, the color key also needs to be annotated. Is it the gene expression level?

Point-by-point response to reviewer comments

We would like to thank all the reviewers again for their comments to improve our manuscript. According to the comments, we have revised the manuscript: text, figure, and supplementary figure files. Our revised text file is a marked-up version, with changes in the text highlighted in blue. The revised manuscript text is also exemplified in our response to each comment.

REVIEWER COMMENTS

Reviewer #1 (Remarks to the Author):

I am pleased to see that the authors have made the changes and addressed my previous comments. So in principle for me the manuscript can be published. Just one last small improvement should be made: I understand the authors have decided to tone down the conclusions of Myc and Gal3 as opposed obtain an in vivo validation. However, I feel that at least a validation of the presence of Gal3 protein in vivo in the metastasis area vs distal liver is needed as a quantification. Fig 4c, shows Gal3 expression in the RNA seq data and Fig 4d shows a representative IF staining, however, this should be shown alongside a quantification of this IF staining from different imaging, which properly can confirm the RNA data, before moving to Fig 4e-f where Gal3 is tested on cancer cells ex vivo.

Also the following sentence in the discussion is unclear “In addition, Cherry-niche has not been detailed for its fluorescent labelling efficiency and applicability in various cell type”. The use of that model was published using breast cancer but also leukaemia cells (Passaro et al., Cell Rep. 2021), wouldn't this show it in very different cells, or the author means something else by this? The sentence is not clear.

Answer: We agree that quantitative analysis of Galectin-3 signals in immunostaining images improves the integrity of our manuscript. We have added the data in the revised manuscript (Fig.S15c).

We appreciate the reviewer's comment pointing out the reference to Cherry-niche application as we hadn't noticed that work. We have therefore removed the discussion for the lack of the Cherry-niche system application and revised the sentence (line 319).

(lines 316-321)

Although the *in vivo* comparison between Cherry-niche and sGRAPHIC is still lacking in this study, we reproduced Cherry-niche labeling with HEK293T cells, and showed that the fluorescent labeling efficiency of sGRAPHIC surpassed that of Cherry-niche (Figs. 1c - e, and S4). ~~In addition, Cherry-niche has not been detailed for its fluorescent labeling efficiency and applicability in various cell types.~~ ~~In contrast,~~ **In addition,** sGRAPHIC archived fluorescence labeling of multiple types of cell-cell interactions with high efficiency (Figs. 2a, b, S5, and S6).

Reviewer #2 (Remarks to the Author):

After conducting a thorough examination of the revised version of the manuscript, I am pleased to note that the authors have comprehensively addressed the comments provided by all the reviewers, including mine. The authors' responsiveness and attention to the reviewers' feedback are commendable, as they have taken the necessary steps to ensure the manuscript's improved quality and validity.

Notably, the authors not only revised the text according to the reviewers' suggestions but also made significant improvements by incorporating additional figures. These additions have improved the visual representation of the research findings, enhancing the overall clarity of the manuscript.

However, one area that still requires attention is the graphical abstract. It still falls short of delivering a clear and concise message about the research. A graphical abstract serves as a visual summary of the study's key findings and implications, aiming to capture the reader's attention and provide a quick overview of the research content. Therefore, I suggest further refining the graphical abstract to ensure that it effectively communicates the main highlights of the research.

Overall, the manuscript's revisions and additions demonstrate the authors' diligence in addressing the reviewers' comments and improving the overall quality of the study.

Answer: We have revised the graphical abstract to visually emphasize the highlight of the research. We believe that this revision improves the visual communication for a wide range of readers.

Reviewer #3 (Remarks to the Author):

Generally, all of my comments have been well-addressed. I have the following minor points to be

further clarified.

LogFC from DE has been used as one of the several criteria in a few figures/results.

Specifically, in Line 939, Figure 4b and Figure S14, is “log2 FC” or “absolute value of log2 FC” used as the criteria?

For Figure 4b, the color key also needs to be annotated. Is it the gene expression level?

Answer: We used $\log_2 \text{FC} > 0.25$ (default) and `only.pos = True` options in *FindAllMarker* function in finding the cluster markers. We have revised the sentences to clarify this point (lines 718 and 725). We have also annotated the color key as gene expression level in Fig.4b and revised the figure caption (line 938).

(lines 718-719)

The counts were normalized with a *LogNormalize* and *scaled* with a scale factor of 100,000.

(lines 723-725)

The marker genes of clustered populations were identified by comparing the cells in a population with all other cells with the *FindAllMarkers* function with the Wilcoxon Rank Sum test and `only.pos = T` option in Seurat (Figs. 4a, 4b and S14).

(lines 938-940)

(b) Heatmap displaying expression level of cluster 2 and 3 marker genes of the top 20 ranked by Log₂ FC (adjusted P-value < 0.05) expressions in the GFP+ proximal hepatocytes.

REVIEWERS' COMMENTS

Reviewer #1 (Remarks to the Author):

The authors have now address my comments and in my opinion the manuscript is ready for publication. On reflection I have a final comment on the title that I did not consider before. Apology for not noticing earlier, it is only a suggestion.

When saying "Genetic optical labeling of neighboring cells interrogates cell–cell interactions in metastatic niches." it sounds like the labelling is a genetic tracing where neighboring cells are genetically induced to express a GFP gene. In this case I believe the "genetic" refer to the fact that the neighboring cells need to be genetically modified in order to be labelled with this system, but it is not really clear before reading the manuscript.

I think it could be misleading and I would suggest to maybe remove the "Genetic" and refer to the system instead.

Something like this:

"Split green fluorescent protein labelling of neighboring cells interrogates cell–cell interactions in metastatic niches".

Something on this line, just to avoid misunderstanding.

Point-by-point response to reviewer comments

We would like to thank the reviewer again for his/her comment to improve our manuscript. According to the comment, we have revised the manuscript and exemplified the revised texts in our response to the comment. Our revised text file is a marked-up version, with changes in the text highlighted in blue. These changes have been made in response to the comments from both the reviewer and the editor.

- - - - -

REVIEWER COMMENT

Reviewer #1 (Remarks to the Author):

The authors have now address my comments and in my opinion the manuscript is ready for publication. On reflection I have a final comment on the title that I did not consider before. Apology for not noticing earlier, it is only a suggestion.

When saying "Genetic optical labeling of neighboring cells interrogates cell–cell interactions in metastatic niches." it sounds like the labelling is a genetic tracing where neighboring cells are genetically induced to express a GFP gene. In this case I believe the "genetic" refer to the fact that the neighboring cells need to be genetically modified in order to be labelled with this system, but it is not really clear before reading the manuscript.

I think it could be misleading and I would suggest to maybe remove the "Genetic" and refer to the system instead.

Something like this:

"Split green fluorescent protein labelling of neighboring cells interrogates cell–cell interactions in metastatic niches".

Something on this line, just to avoid misunderstanding.

Answer: We agree that the previous title may mislead some readers in certain fields. As a new title, we would like to emphasize our novelty “secretory” in combination with split GFP system. Also, we

prefer to use the term of “GFP reconstitution” to describe the GFP fragmentation system in the historical context of GRASP and GRAPHIC. We believe that the new title “Secretory GFP reconstitution labeling of neighboring cells interrogates cell–cell interactions in metastatic niches” would correctly convey the concept of our system to most readers. Along with the new title, the word of “genetic optical labeling” in the manuscript has been revised.

(line 83)

however, it remains challenging to develop **genetically encoded** optical-labeling tools for harnessing high labeling

(line 268)

Here, we demonstrated that our **secretory GFP reconstitution** labeling system, sGRAPHIC, is a powerful tool for selectively labeling tissue-resident cells neighboring on cancer cells in co-culture systems and in murine models.

(Figure 1 legend title)

Development of a system for **secretory GFP reconstitution labeling of neighboring cells.**